# Automated multilabel diagnosis on electrocardiographic images and signals

Veer Sangha[1], Bobak J. Mortazavi [2,3], Adrian D. Haimovich [4], Antônio H. Ribeiro[5], Cynthia A. Brandt[4,6], Daniel L. Jacoby[7], Wade L. Schulz [3,8], Harlan M. Krumholz [3,7,9], Antonio Luiz P. Ribeiro [10,11] & Rohan Khera [3,7✉]

The application of artificial intelligence (AI) for automated diagnosis of electrocardiograms (ECGs) can improve care in remote settings but is limited by the reliance on infrequently available signal-based data. We report the development of a multilabel automated diagnosis model for electrocardiographic images, more suitable for broader use. A total of 2,228,236 12-lead ECGs signals from 811 municipalities in Brazil are transformed to ECG images in varying lead conformations to train a convolutional neural network (CNN) identifying 6 physician-defined clinical labels spanning rhythm and conduction disorders, and a hidden label for gender. The image-based model performs well on a distinct test set validated by at least two cardiologists (average AUROC 0.99, AUPRC 0.86), an external validation set of 21,785 ECGs from Germany (average AUROC 0.97, AUPRC 0.73), and printed ECGs, with performance superior to signal-based models, and learning clinically relevant cues based on Grad-CAM. The model allows the application of AI to ECGs across broad settings.

[1] Department of Computer Science, Yale University, New Haven, CT, USA. [2] Department of Computer Science & Engineering, Texas A&M University, College Station, TX, USA. [3] Center for Outcomes Research and Evaluation, Yale-New Haven Hospital, New Haven, CT, USA. [4] Department of Emergency Medicine, Yale University School of Medicine, New Haven, CT, USA. [5] Department of Information Technology, Uppsala University, Uppsala, Sweden. [6] VA Connecticut Healthcare System, West Haven, CT, USA. [7] Section of Cardiovascular Medicine, Department of Internal Medicine, Yale School of Medicine, New Haven, CT, USA. [8] Department of Laboratory Medicine, Yale School of Medicine, New Haven, CT, USA. [9] Department of Health Policy and Management, Yale School of Public Health, New Haven, CT, USA. [10] Telehealth Center and Cardiology Service, Hospital das Clínicas, São Paulo, Brazil. [11] Department of Internal Medicine, Faculdade de Medicina, Universidade Federal de Minas Gerais, Belo Horizonte, Brazil. ✉email: rohan.khera@yale.edu

Electrocardiography is an essential tool in the diagnosis and management of cardiovascular diseases, serving as an avenue for the identification of key pathophysiological signatures from the electrical activity of the heart. Currently, data from electrocardiograms (ECGs) are collected as multichannel surface signal recordings of the cardiac electrical activity that are then transformed to images with printed waveforms. These images are then interpreted by trained clinicians, often precluding immediate diagnosis or the use of technology that can deliver deeper insights. While the automated interpretation of ECGs promises to improve clinical workflow, particularly for key cardiovascular conditions, these tools are based on raw electrocardiographic signals[1,2] rather than printed images.

Specifically, deep learning has been applied successfully to automate diagnosis based on signal data, performing comparably to trained clinicians for tasks such as the detection of ECG abnormalities[1–3]. However, a reliance on signal-based models poses a challenge in the real-world application of automated diagnosis, as ECGs are frequently printed and scanned as images. Thus, a major reorganization of operation is required to facilitate the application of models that focus on signals. Such technology is also inaccessible to paraclinical staff serving in remote settings, or to patients who increasingly have access to electrocardiographic images but lack ready access to experts for early diagnosis. Few tools have focused on the automated diagnosis that allows for the incorporation of both ECG images and signals. Many existing models are trained and tested on data from a single source, with an inability to infer broad generalizability to different institutions and health settings. Finally, there has also been a preponderance of tools focusing on the diagnosis of single clinical entities[4,5], limiting clinical utility as ECGs can have multiple abnormalities simultaneously.

We developed a multilabel prediction algorithm that can incorporate either ECG images or signals as inputs to predict the probability of various rhythm and conduction disorders using over 2 million ECGs from Brazil, independently validated in data from Germany, and for the image-based model, on real-world printed ECGs.

## Results

**Study population.** We used 12-lead ECGs collected by the Telehealth Network of Minas Gerais (TNMG) and described previously in Ribeiro et al.[3]. The data were assembled as a part of the clinical outcomes in digital electrocardiography (CODE) study[6]. There were 2,228,236 ECGs from 1,506,112 patients acquired between 2010 and 2017 from 811 out of the 853 municipalities in the state of Minas Gerais, Brazil. The median age of the patients at the time of the ECG recording was 54 years (IQR 41, 67) and 60.3% of the ECGs were obtained among women. These ECGs were recorded as standard 12-lead recordings sampled at frequencies ranging from 300 to 600 Hz for 7– 10 s. In addition, information on patient demographics and 6 clinical labels were available (See "Methods" for details).

Of these ECGs, 39,661 (1.8%) had a label for atrial fibrillation (AF), 61,551 (2.8%) for right bundle branch block (RBBB), 34,677 (1.6%) for left bundle branch block (LBBB), 34,446 (1.5%) for first-degree atrioventricular block (1dAVb), 35,441 (1.6%) for sinus bradycardia (SB), and 48,296 (2.2%) for sinus tachycardia (ST) (Supplementary Table 1). Of the 231,704 ECGs with at least one of the six detected rhythm disorders, 210,496 had exactly one, and 21,208 had more than one label.

Further, to augment the evaluation of models built on the primary CODE study dataset, where clinical annotations were derived from routine clinical care and therefore a single clinician, a secondary cardiologist-validated annotation test dataset was used. This consisted of 827 additional ECGs obtained from the TNMG network between April and September 2018, with a similar distribution of age, sex, and clinical labels (Supplementary Table 1). These ECGs were rigorously validated by 2-to-3 independent cardiologists based on criteria from the American Heart Association[7]. An ECG dataset from Germany was obtained for external validation (described in Methods, external validation).

In addition to these signal waveform databases that had been transformed to images, image models were evaluated on two real-world image datasets. The first of these was from a rural US hospital system, and the second consisted of web-based ECG images (described in Methods, external validation).

**Performance of image-based multilabel classification.** We resampled all ECGs to a 300 Hz sampling rate with 5 s simultaneous recordings across leads to obtain uniform length inputs. QRS peak detection by the XQRS algorithm in lead I of the original recording aided in the determination of the start and end of the recordings to be assessed with the signal models and to be converted to ECG images[8]. ECGs that did not have any peaks detected (94,277, or 4.1%) were discarded, as inspection of a sample of 50 ECGs after further preprocessing did not detect any cardiac activity but rather uninterpretable noise. The ECGs were transformed to the corresponding printed images using the python library ecg-plot[9].

All ECGs were randomly subset into training, validation, and test sets (90%-5%-5%). Given the low prevalence of all clinical labels, the data splits were stratified by clinical labels, so cases of each of the six labeled clinical disorders were split proportionally among the sets, as was gender. From the training waveforms, two image-based datasets were created for image model training, including a standardized subset, and a real-world variation subset.

In the standardized subset, each pre-image transformation sample started 300 ms before the second QRS peak detected in lead I. To ensure that like a clinician the model learned about lead-specific information based on the label of the lead, rather than just the location of leads on a single ECG format, we included four formats of images in the standardized subset (Fig. 1). Standard printed ECGs in the United States typically consist of four 2.5 second columns presented sequentially on the page, with each column containing three leads of 2.5 s intervals of a continuous 10 s record. A 10-s rhythm strip was generated using a concatenation of lead I signal on the last and first QRS of the same 5 s signal[7]. The second plotting scheme we chose consisted of two columns each containing six 5-s recordings, with one column containing simultaneous limb leads and the other simultaneous precordial leads. We treated this scheme as our alternate image form (Fig. 1). The third and fourth formats were representations of the standard and alternate formats with shuffled lead locations. In the shuffled standard format precordial leads were presented in the first two columns and limb leads in the third and fourth. In shuffled alternative images precordial leads were presented in the first column and limb leads in the second (Fig. 1). This was done to prevent overfitting to certain locations of ECG leads on the image, and to create a more versatile algorithm.

The second real-world variation subset of the training sample included images based on the same waveform signals as the standardized subset. All ECGs were in the standard layout of the standardized subset, but variations expected in real-world ECG images were introduced in constructing these additional images. These variations fell into two categories. First, each pre-image transformation sample started at a point that was selected from a uniform distribution of 0–1.3 s offset from the first detected QRS

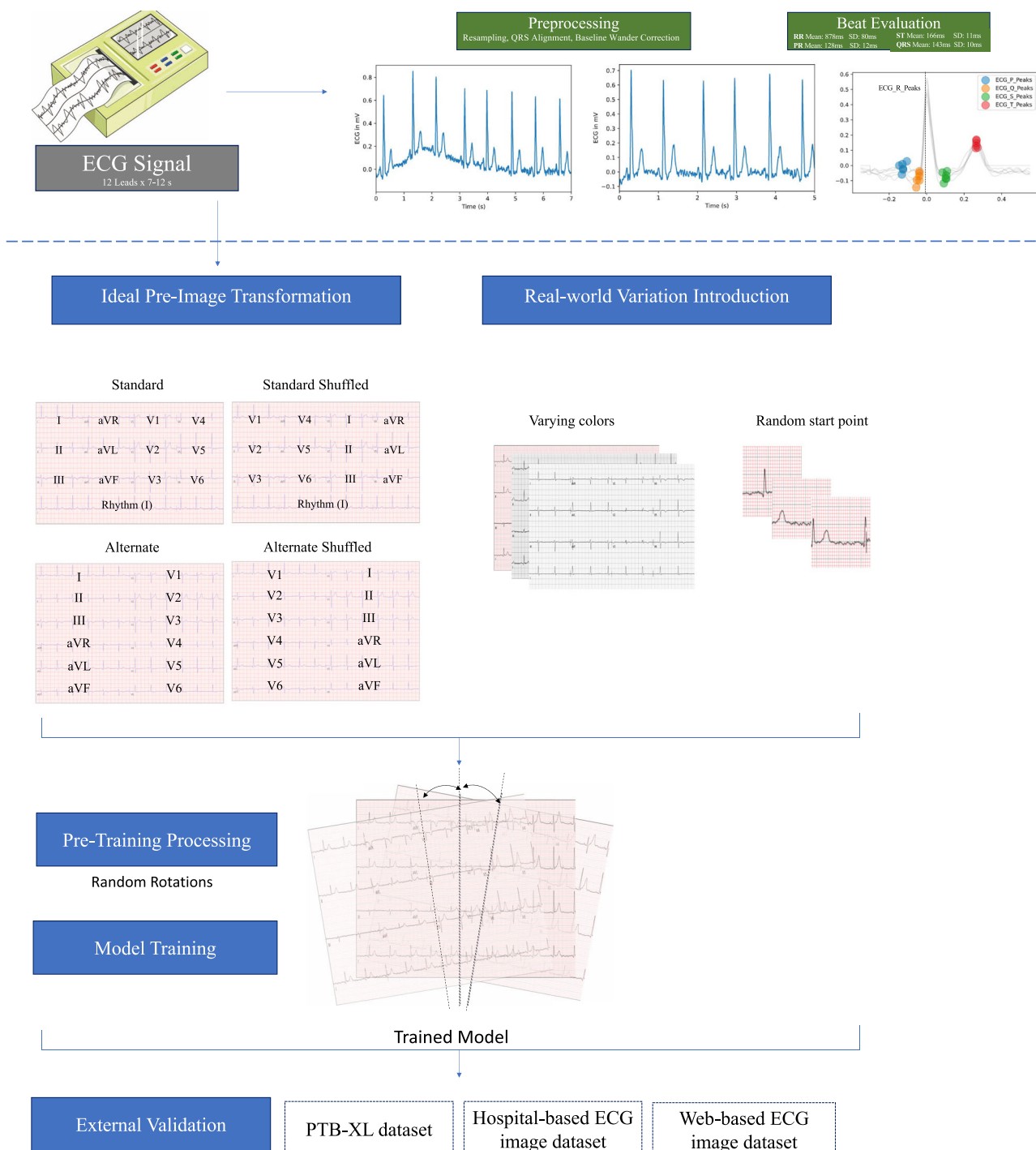

**Fig. 1 Study outline for waveform preprocessing and image transformation for modeling.** ECG electrocardiogram.

peak in lead I. This was designed to mimic the lack of predetermined starting points in real-world ECGs. Second, color schemes for the background of the ECG image plotted from the signals were varied across ECGs (Details in "Methods") (Fig. 1).

For image data, we built a convolutional neural network (CNN) model based on the EfficientNet B3 architecture (see "Methods" for details)[10]. We trained our primary image model on both the standardized subset, with a 40%-40%-10%-10% split of standard, alternate, standard shuffled, and alternate shuffled images as well as the real-world variation subset. Validation was conducted on standardized images. All images were randomly rotated between −10 and +10 degrees prior to being input into

the model for training and validation to mimic variation that might be seen in uploaded ECGs and further aid in the prevention of overfitting.

To ensure that model learning was not affected by the low frequency of certain labels, custom loss functions based on the effective number of samples class sampling scheme were used for both image and signal models, with weighting based on the number of samples for each class (Supplementary Table 2)[11].

The AUROCs for clinical labels on the held-out test set ranged from 0.98 for 1dAVb to 1.00 (rounded up from 0.997) for LBBB (Table 1). AUROC was 0.99 or higher for RBBB, LBBB, SB, AF, and ST, but was lower for 1dAVB. The AUPRCs for clinical labels

**Table 1 Performance of image and signal based models on held-out test set and cardiologist validated test set.**

| Data source | Model | Label | Accuracy | PPV | NPV | Specificity | Sensitivity | AUROC | F1 | AUPRC |
|---|---|---|---|---|---|---|---|---|---|---|
| Held-out test set | Image | Male | 0.865 | 0.817 | 0.898 | 0.875 | 0.849 | 0.934 (0.933–0.936) | 0.833 | 0.905 |
| | | 1dAVb | 0.985 | 0.499 | 0.994 | 0.990 | 0.605 | 0.982 (0.980–0.984) | 0.547 | 0.509 |
| | | RBBB | 0.989 | 0.750 | 0.997 | 0.992 | 0.878 | 0.994 (0.994–0.995) | 0.809 | 0.801 |
| | | LBBB | 0.994 | 0.765 | 0.998 | 0.996 | 0.854 | 0.997 (0.997–0.997) | 0.807 | 0.826 |
| | | SB | 0.986 | 0.537 | 0.997 | 0.988 | 0.818 | 0.991 (0.989–0.992) | 0.648 | 0.594 |
| | | AF | 0.993 | 0.778 | 0.997 | 0.996 | 0.826 | 0.994 (0.993–0.995) | 0.801 | 0.822 |
| | | ST | 0.987 | 0.645 | 0.997 | 0.990 | 0.859 | 0.994 (0.993–0.995) | 0.737 | 0.728 |
| | | Weighted mean | 0.989 | 0.672 | 0.997 | 0.992 | 0.817 | 0.992 (0.991–0.993) | 0.735 | 0.725 |
| | Signal | Male | 0.704 | 0.594 | 0.829 | 0.641 | 0.800 | 0.803 (0.801–0.806) | 0.682 | 0.737 |
| | | 1dAVb | 0.968 | 0.214 | 0.991 | 0.976 | 0.411 | 0.934 (0.930–0.938) | 0.282 | 0.197 |
| | | RBBB | 0.987 | 0.733 | 0.995 | 0.992 | 0.827 | 0.987 (0.986–0.989) | 0.777 | 0.745 |
| | | LBBB | 0.992 | 0.697 | 0.997 | 0.994 | 0.821 | 0.994 (0.993–0.995) | 0.754 | 0.754 |
| | | SB | 0.986 | 0.561 | 0.996 | 0.990 | 0.767 | 0.990 (0.990–0.991) | 0.648 | 0.589 |
| | | AF | 0.984 | 0.535 | 0.994 | 0.990 | 0.643 | 0.979 (0.976–0.982) | 0.584 | 0.567 |
| | | ST | 0.986 | 0.639 | 0.997 | 0.989 | 0.854 | 0.992 (0.991–0.994) | 0.731 | 0.708 |
| | | Weighted mean | 0.984 | 0.584 | 0.995 | 0.989 | 0.738 | 0.981 (0.979–0.983) | 0.649 | 0.615 |
| Cardiologist-validated test set | Image | Male | 0.815 | 0.744 | 0.865 | 0.826 | 0.798 | 0.890 (0.867–0.912) | 0.770 | 0.845 |
| | | 1dAVb | 0.985 | 0.711 | 0.999 | 0.986 | 0.964 | 0.995 (0.991–0.999) | 0.818 | 0.866 |
| | | RBBB | 0.989 | 0.903 | 0.992 | 0.996 | 0.824 | 0.995 (0.992–0.999) | 0.862 | 0.897 |
| | | LBBB | 1.000 | 1.000 | 1.000 | 1.000 | 1.000 | 1.000 (1.000–1.000) | 1.000 | 1.000 |
| | | SB | 0.995 | 0.800 | 1.000 | 0.995 | 1.000 | 0.997 (0.994–1.000) | 0.889 | 0.797 |
| | | AF | 0.995 | 0.800 | 0.999 | 0.996 | 0.923 | 0.997 (0.993–1.000) | 0.857 | 0.881 |
| | | ST | 0.992 | 0.895 | 0.996 | 0.995 | 0.919 | 0.998 (0.996–1.000) | 0.907 | 0.961 |
| | | Weighted mean | 0.992 | 0.867 | 0.997 | 0.995 | 0.930 | 0.997 (0.994–1.000) | 0.893 | 0.915 |
| | Signal | Male | 0.712 | 0.594 | 0.847 | 0.646 | 0.816 | 0.795 (0.764–0.825) | 0.688 | 0.703 |
| | | 1dAVb | 0.970 | 0.538 | 0.991 | 0.977 | 0.750 | 0.970 (0.947–0.993) | 0.627 | 0.607 |
| | | RBBB | 0.990 | 0.842 | 0.997 | 0.992 | 0.941 | 0.994 (0.989–1.000) | 0.889 | 0.881 |
| | | LBBB | 0.996 | 0.966 | 0.997 | 0.999 | 0.933 | 0.997 (0.993–1.000) | 0.949 | 0.967 |
| | | SB | 0.989 | 0.667 | 0.998 | 0.991 | 0.875 | 0.996 (0.992–1.000) | 0.757 | 0.814 |
| | | AF | 0.990 | 0.667 | 0.996 | 0.994 | 0.769 | 0.986 (0.966–1.000) | 0.714 | 0.746 |
| | | ST | 0.994 | 0.944 | 0.996 | 0.997 | 0.919 | 0.998 (0.996–1.000) | 0.932 | 0.952 |
| | | Weighted mean | 0.988 | 0.803 | 0.996 | 0.992 | 0.880 | 0.991 (0.982–0.999) | 0.836 | 0.848 |

*1dAVB* 1st degree AV block, *AF* atrial fibrillation, *AUPRC* area under precision recall curve, *AUROC* area under receiver operator characteristic curve, *LBBB* left bundle branch block, *NPV* negative predictive value, *PPV* positive predictive value, *RBBB* right bundle branch block, *SB* sinus bradycardia, *ST* sinus tachycardia.

ranged from 0.51 for 1dAVb to 0.83 for LBBB. AUPRC was >0.73 for RBBB, LBBB, AF, and ST, with lower values for SB and 1dAVB (Table 1). At cutoffs that ensured maximum F1 value, specificity was above 0.98 for all clinical labels, and sensitivity was above 0.82 for RBBB, LBBB, SB, AF, and ST, but was lower for 1dAVb (0.61) (Table 1). For the higher-order label of gender, AUROC was 0.93, AUPRC 0.91, and specificity and sensitivity 0.88 and 0.85 respectively.

Performance on standard and alternate format images was comparable. The class mean weighted AUROC across clinical labels on the held-out test set images in both standard and alternate format was 0.99 (Supplementary Table 3). The class mean weighted AUPRC across clinical labels was 0.73 on images for the standard format, and 0.72 for images in the alternate format. For the higher-order label of gender, AUROC was 0.93 for both standard and alternate format images, and AUPRC was 0.91 for standard format images and 0.90 for alternate format images.

To explicitly test the robustness of this model to the noise-artifacts we anticipated in the real world, we tested the model on a representative subset of 5000 ECGs in our held-out test set. We compared model performance on four variations of the same 5000 ECG waveform data. They were plotted in the standard ECG layout, first without additional variation, second, in black and white, third, with random rotations between −10 and +10 degrees, and fourth, with a varying starting point for plotted ECG with a 0–1.3 s offset from the first detected QRS complex in the signal. We saw no drop off in the performance of the model as various types of noise were added (Supplementary Table 3). The model is available as a publicly accessible web-based tool[12].

**Performance of signal-based multilabel classification**. For signal data, we developed a custom CNN model that combined inception blocks, convolutional, and fully connected layers, similar to Raghunath et al. (see "Methods" for details)[4]. The AUROCs for clinical labels on the held-out test set ranged from 0.93 for 1dAVb to 0.99 for LBBB. AUROC was >0.98 for RBBB, LBBB, SB, AF, and ST, but was lower for 1dAVB (Table 1). The AUPRCs for clinical labels ranged from 0.20 for 1dAVb to 0.75 for LBBB. AUPRC was >0.70 for RBBB, LBBB, and ST, and >0.55 for SB and AF. At cutoffs that corresponded to the highest F1 value, specificity was above 0.98 for all clinical labels, and sensitivity was above 0.75 for RBBB, LBBB, SB, AF, and ST, but was lower for 1dAVb (0.41). For the higher-order label of gender, AUROC was 0.80, AUPRC 0.74, and specificity and sensitivity 0.64 and 0.80 respectively.

**Internal validation**. In addition to the held-out test set, we validated the algorithm in an expert validated test of 827 ECGs from distinct patients in Minas Gerais, Brazil during a different time window (April to September 2018). The labels on these ECGs were validated by two cardiologists, with disagreements resolved by a third cardiologist, based on criteria by the American Heart Association.

The class weighted mean AUROC across clinical labels on the cardiologist-validated internal test set was higher than the held-out test. For the image-based model, it was 1.00 (rounded up from actual value, 0.997) (95% CI, 0.99–1.00), and for the signal-based model it was 0.99 (95% CI, 0.98–1.00) (Table 1). Class weighted mean AUPRCs were also higher on the cardiologist-validated test set, 0.92 for the image-based model and 0.85 for the signal-based model. Performance on the higher-order label of gender was slightly lower for the image-based model, with an AUROC of 0.89 (95% CI, 0.87–0.91) and AUPRC of 0.85 but was

comparable for the signal-based model (AUROC 0.80 and AUPRC 0.70).

**External validation: ECG waveform database**. In addition to the held-out test set, model performance was evaluated on the Germany-based external validation dataset, PTB-XL, whose data have been previously described[13]. Briefly, the dataset has 21,837 recordings from 18,885 patients. ECGs were collected with devices from Schiller AG between 1989 and 1996 and are available as 10 s recordings sampled at 500 Hz for 10 s. Each record has labels for diagnostic, form, and rhythm statements. Data were transformed in the same manner as the data from Brazil, and the same labels were extracted to assess model performance. The model performance in PTB-XL was comparable to the held-out test set. For the image-based model, class weighted mean AUROC was 0.98 (95% CI, 0.98–0.98) and for the signal-based model, it was 0.96 (95% CI, 0.95–0.96) (Table 2). Class weighted mean AUPRCs were also comparable, 0.78 for the image-based model and 0.65 for the signal-based model. Performance on the higher-order label gender was also comparable across datasets. For the image-based model, AUROC was 0.90 (95% CI, 0.90–0.90) and AUPRC was 0.90 on PTB-XL, and for the signal-based model, AUROC was 0.74, and AUPRC was 0.75.

**External validation: real-world ECG images**. We also pursued validation on two real-world image datasets. The first of these was from the Lake Regional Hospital System (LRH), a rural US hospital in Osage Beach, MO. These data included 64 ECG images including 8–10 ECGs for our six labels of interest, as well as ECGs that were labeled as normal. Subjectively, the ECGs had a similar layout as the standard ECG format but had the V1 lead rather than lead I as the rhythm strip (a single lead with a full 10 s recording for identifying rhythm). There were vertical lines demarcating the leads, the signal was black rather than blue, and the background color and grid of the ECGs varied, as did the location and the font of lead label. Model performance on the LRH dataset was also comparable to the held-out test set. Class weighted mean AUROC and AUPRC were 0.98 and 0.94, respectively (Table 3). For gender, the model had an AUROC and AUPRC of 0.78.

The second real-world image dataset consisted of ECG images available on the internet, representing 42 ECGs. The approach to obtaining these images is outlined in the methods. All ECG labels were confirmed by a cardiologist. Qualitatively, these web ECGs included both standard and alternate format images, as well as new image formats such as ones with no rhythm leads, or two or more rhythm leads. Moreover, the ECGs had varying background colors, signal colors, quality, and location of lead labels, and many had additional artifacts that were not present in our training data. These images are available upon request. The model achieved good performance on this web-based dataset, with class weighted mean AUROC and AUPRC of 0.93 and 0.80 respectively, and a high discrimination across labels (Table 3).

**Comparison of performance of image and signal models**. The image and signal models performed comparably for clinical labels on both datasets, with high correlation between prediction across labels. For the clinically important diagnosis of AF, the image-based model had AUROCs of 0.99 (95% CI, 0.99–1.00) on the held-out test set, 1.00 (95% CI, 0.99–1.00) on the cardiologist-validated internal test, and 0.99 (95% CI, 0.99–1.00) on PTB-XL, while the signal-based model had AUROCs of 0.98 (95% CI, 0.98–0.98), 0.99 (95% CI, 0.97–1.00) and 0.97 (95% CI, 0.97–0.98) respectively.

The class weighted mean AUROC across clinical labels was also comparable; 0.99 (95% CI, 0.99–0.99) on the held-out test set, 1.00 (95% CI, 0.99–1.00) on the cardiologist-validated internal test set, and 0.98 (95% CI, 0.98–0.98) on PTB-XL for the image-based model, and 0.98 (95% CI, 0.98–0.98), 0.99 (95% CI, 0.98–1.00), and 0.96 (95% CI, 0.95–0.96) for the signal-based model (Tables 1, 2). For the higher-order label of gender, the image-based model outperformed the signal-based model, with AUROC of 0.93, 0.89, and 0.90 on the held-out test, cardiologist-validated test, and PTB-XL, respectively, compared with 0.80, 0.80, and 0.74 for the signal-based model ($p < 0.001$ for difference on held-out test, PTB-XL). The high discrimination across labels and in all three datasets for both image and signal-based models was noted in ROC curves (Figs. 2, 3).

The label-level performance of image and signal based models was also consistent, with the highest AUROC and AUPRC scores on the same clinical labels, LBBB and RBBB, (AUROC of 1.00 (0.997) and 0.99 for image-based, and 0.99 and 0.99 for signal based, and AUPRC of 0.83 and 0.80 for image-based and 0.75 and 0.75 for signal-based on the held-out test set) and lowest scores on the same class, 1dAVb (AUROC of 0.98 for image-based, and 0.93 for signal based, and AUPRC of 0.51 for image-based, and 0.20 for signal-based on the held-out test set). Confusion matrices showed that among ECGs with only one clinical label, predictions of LBBB, RBBB, and ST were the most accurate for both image and signal-based models (above 87% for all three for the image-based model, and above 86% for the signal-based one) (Fig. 4, Supplementary Fig. 3). These findings were consistent in the cardiologist-validated set and PTB-XL.

**Manual review for misclassified ECGs**. Two cardiologists independently reviewed a sample of 10 false positives for each clinical label with the highest probability of a given label ($n = 120$) to verify the accuracy of the labels and qualitatively assess the potential ECG features that may have prompted a false positive result in both the held-out test set and the external validation data. The most common errors across algorithms were type 1, or false positives. We took 120 false positives from our image-based model with the highest probability for each clinical label in the held-out test set and external validation data, PTB-XL (10 for each label in each dataset). Expert review by cardiologists confirmed that all 120 were accurately classified by the model and had incorrect labels, i.e., that these were true positive results (ECGs available at https://github.com/CarDS-Yale/ECG-DualNet).

**Prediction interpretability with Grad-CAM**. We used Gradient-weighted Class Activation Mapping (Grad-CAM) to highlight the regions in an image predicting a given label[14]. This provides interpretability of the model's predictions and allows for the evaluation of whether model-assigned labels are based on clinically relevant information or on heuristics based on spurious data features[15]. The Grad-CAMs identified sections of the ECG that were most important for the label classification. Figure 5 shows the average class activation heatmaps for RBBB and LBBB predictions on standard and alternate form images, highlighting ECG regions that were important for the diagnosis of each rhythm across many predictions. We chose LBBB and RBBB to illustrate the interpretability as these labels have lead-specific information that is used to make the clinical diagnosis. This contrasts with the other labels (AF, ST, SB, 1dAVB) where the information on these rhythm disorders can be deduced from any of the ECG leads, limiting assessment of lead-specific learning.

The region of the ECG corresponding to the precordial leads was the most important for the prediction of RBBB across both the standard and alternate images, with the region corresponding to leads V4 and V5 especially important in the standard format, and V1, V2, and V3 in the alternate format. On the other hand, while regions corresponding to lead V6 was most important for the prediction of LBBB across standard images, regions corresponding to lead V4 and V5 were most important for LBBB predictions across alternate images. Both formats showed significant attention across precordial leads. The rhythm lead was also important for the prediction of both LBBB and RBBB in the standard format.

Supplementary Fig. 4 shows Grad-CAMs for individual representative examples of model prediction of RBBB and LBBB on real-world images from the web-based dataset. In both examples of RBBB, the region corresponding to leads V1 and V2 is most important for the prediction of the label. In the two examples of LBBB, precordial leads are again the most important for the prediction of the label, despite varying in the relative position of the leads and the difference in the number and type of the continuous rhythm strip at the bottom of the ECG image.

**Sensitivity analyses**. Our image-based model performed comparably on both standard and alternate form printed images in the held-out test set and cardiologist validated test set (Supplementary Tables 4 and 5).

In the second sensitivity analysis, we trained a signal model with the exact same architecture as the one described, but without peak morphology inputs to test the ability of a convolutional network to perform these operations internally and learn the same patterns. Our signal model without peak morphology information also performed comparably to the signal model with them (Supplementary Tables 6 and 7).

## Discussion

We have developed an externally validated multilabel automated diagnosis algorithm that accurately identifies rhythm and conduction disorders from either ECG images or raw electrocardiographic signals. The algorithm demonstrates high discrimination and generalizes across two international waveform data sources, which are acquired on different equipment and temporally separated by over 2 decades. The image-based models are also invariant to the layout of the ECG images, with interpretable recognition of leads of interest and abnormalities by the image-based algorithms. The model also demonstrated high discrimination of clinical labels on two real-world image datasets, with varying ECG layouts as well as additional artefacts. The model also demonstrates consistent performance in identifying the gender of patients from ECGs, highlighting that ECG images, like signals, can be used to identify hidden features, which has thus far exclusively been done with signal-based models. Our approach has the potential to broaden the application of artificial intelligence (AI) to electrocardiographic data across storage strategies.

Image-based models represent an important advance in automated diagnosis from ECGs as they allow applications to data sources for which raw signals may not be available. This represents most healthcare settings that have not been optimized for storing and processing signal data in real-time and that rely on printed or scanned ECG images. In addition, in contrast to ECG-based models that have been developed in single health systems[2,16,17], our models have broad external validity, performing equivalently in regionally and temporally distinct validation data. An important observation is that image-based models demonstrate comparable performance to our signal-based model, as well as signal-based models in published reports[2,18] despite both the substantial downsampling of high-frequency signal recordings to relatively low-resolution images and the

**Table 2 Performance of Image and Signal Based Models on the external validation set PTB-XL.**

| Model | Label | Accuracy | PPV | NPV | Specificity | Sensitivity | AUROC | F1 | AUPRC |
|---|---|---|---|---|---|---|---|---|---|
| Image | Male | 0.810 | 0.784 | 0.845 | 0.738 | 0.876 | 0.899 (0.895–0.903) | 0.827 | 0.904 |
| | 1dAVb | 0.960 | 0.457 | 0.984 | 0.974 | 0.573 | 0.946 (0.940–0.953) | 0.508 | 0.457 |
| | RBBB | 0.989 | 0.712 | 0.998 | 0.990 | 0.939 | 0.996 (0.995–0.997) | 0.81 | 0.793 |
| | LBBB | 0.993 | 0.823 | 0.998 | 0.995 | 0.898 | 0.997 (0.995–0.998) | 0.859 | 0.902 |
| | SB | 0.980 | 0.673 | 0.989 | 0.991 | 0.641 | 0.954 (0.945–0.963) | 0.656 | 0.641 |
| | AF | 0.986 | 0.898 | 0.992 | 0.992 | 0.898 | 0.993 (0.990–0.995) | 0.898 | 0.934 |
| | ST | 0.984 | 0.759 | 0.994 | 0.990 | 0.838 | 0.991 (0.989–0.993) | 0.796 | 0.809 |
| | Weighted mean | 0.982 | 0.743 | 0.992 | 0.989 | 0.805 | 0.981 (0.977–0.984) | 0.770 | 0.776 |
| Signal | Male | 0.651 | 0.615 | 0.759 | 0.398 | 0.883 | 0.740 (0.733–0.746) | 0.725 | 0.748 |
| | 1dAVb | 0.919 | 0.190 | 0.976 | 0.939 | 0.377 | 0.862 (0.852–0.872) | 0.253 | 0.177 |
| | RBBB | 0.985 | 0.660 | 0.996 | 0.989 | 0.847 | 0.986 (0.981–0.992) | 0.742 | 0.729 |
| | LBBB | 0.991 | 0.819 | 0.996 | 0.996 | 0.824 | 0.995 (0.993–0.997) | 0.821 | 0.87 |
| | SB | 0.980 | 0.659 | 0.990 | 0.989 | 0.680 | 0.955 (0.947–0.963) | 0.669 | 0.642 |
| | AF | 0.957 | 0.676 | 0.981 | 0.974 | 0.742 | 0.972 (0.969–0.975) | 0.707 | 0.729 |
| | ST | 0.982 | 0.745 | 0.991 | 0.990 | 0.779 | 0.987 (0.983–0.990) | 0.762 | 0.794 |
| | Weighted mean | 0.965 | 0.619 | 0.986 | 0.977 | 0.701 | 0.958 (0.953–0.963) | 0.653 | 0.653 |

*1dAVB 1st degree AV block, AF atrial fibrillation, AUPRC area under precision recall curve, AUROC area under receiver operator characteristic curve, LBBB left bundle branch block, NPV negative predictive value, PPV positive predictive value, RBBB right bundle branch block, SB sinus bradycardia, ST sinus tachycardia.*

redundant information introduced by the presence of background pixels. We do not have a definitive explanation of these observations, though pixel-level information that is not interpretable by humans may include more detailed diagnostic clues than the review of broad waveform patterns used by human readers. Moreover, it is possible that while the signal-based models have a higher frequency record of the electrocardiographic activity of the heart, and therefore, more data points that there is spatial information outside of waveform data that is better represented in printed images, or that the additional high-frequency recording of noisier signal data does not necessarily have more information than image data. We cannot definitively prove if either of these is the case.

These models had an excellent performance on a validation dataset that was manually annotated and confirmed by cardiologists. The reported performance on the held-out test and external validation sets were limited by the quality of labels, which likely varied given higher performance on LBBB, RBBB, and AF, than for 1st degree AV block, a pattern was also observed in signal-based models[3]. We confirmed this explicitly through expert ECG review of a sample of reportedly false-positive ECGs —those where the model predicted a label with high confidence but the ECG was not actually labeled with the condition. We found that these were in fact true positives, representing incorrect labels.

We also found that the model performance was not limited to images generated from the waveform data but extended to those obtained as printed ECGs directly from different sources. The included ECGs spanned different colors, lead positions, and extraneous artifacts noted on a subjective review of these data, but the model continued to have high discrimination, precision, and recall. These observations suggest the utility of both the real-world application of our model, but also provide a strategy for training ECG-based models on ECG images generated from current repositories of ECG signals and labels through the simulated introduction of real-world artifacts in the training data.

Of note, the image-based model learned to identify individual leads on varying ECG formats and identified lead-specific diagnostic cues, similar to human readers. In addition to external validation performance suggesting that the model did not train to rely on dataset-specific heuristics, the interpretability of the AI-based predictions further supports their generalizability[15]. Our examination of mean heatmaps across a sample of predictions for RBBB and LBBB demonstrates that classifications for these intraventricular conduction disturbances were guided by the same information human readers focus on when reading an ECG. Mean heatmaps consistently demonstrated the identification of specific leads that are important in the clinical diagnosis of RBBB and LBBB. Similar heatmaps applied to individual ECGs further supported similar interpretable learning across clinical labels. The identification of specific leads despite the variety of inputs in the training data, and the rotation of training images suggests that the model has learned more generalizable representations of ECG images, especially as it still identified clinically relevant leads in formats of images in the web-based real-world image dataset, which it had never encountered in model training.

Such interpretability addresses some of the implementation challenges of AI-based ECG models that are not readily explainable[19]. Our findings also suggest that Grad-CAM can ameliorate this issue in real-time by providing an automated label, but also informing a clinician visually of the portions of the ECG that were used by the model to ascribe the label. Providing a Grad-CAM output in addition to the diagnosis and confidence of the diagnosis can provide context to predictions made by CNNs and aid in their acceptance in the clinical workflow[1,20].

**Table 3 Performance of image model on real-world electrocardiogram (ECG) image datasets.**

| Real world dataset | Label | Accuracy | PPV | NPV | Specificity | Sensitivity | AUROC | F1 | AUPRC |
|---|---|---|---|---|---|---|---|---|---|
| LRH | Male | 0.719 | 0.667 | 0.842 | 0.516 | 0.909 | 0.778 | 0.769 | 0.784 |
| | 1dAVb | 0.953 | 0.833 | 0.981 | 0.962 | 0.909 | 0.981 | 0.870 | 0.902 |
| | RBBB | 1.000 | 1.000 | 1.000 | 1.000 | 1.000 | 1.000 | 1.000 | 1.000 |
| | LBBB | 0.984 | 1.000 | 0.981 | 1.000 | 0.909 | 0.983 | 0.952 | 0.957 |
| | SB | 0.938 | 0.875 | 0.946 | 0.981 | 0.700 | 0.965 | 0.778 | 0.861 |
| | AF | 0.969 | 0.867 | 1.000 | 0.961 | 1.000 | 0.988 | 0.929 | 0.945 |
| | ST | 0.969 | 0.900 | 0.981 | 0.981 | 0.900 | 0.987 | 0.900 | 0.938 |
| | Weighted mean | 0.969 | 0.912 | 0.983 | 0.980 | 0.909 | 0.984 | 0.908 | 0.935 |
| Web-based | 1dAVb | 0.884 | 0.625 | 0.943 | 0.917 | 0.714 | 0.810 | 0.667 | 0.679 |
| | RBBB | 0.930 | 0.700 | 1.000 | 0.917 | 1.000 | 0.944 | 0.824 | 0.693 |
| | LBBB | 0.930 | 0.833 | 0.946 | 0.972 | 0.714 | 0.897 | 0.769 | 0.763 |
| | SB | 0.977 | 0.875 | 1.000 | 0.972 | 1.000 | 0.992 | 0.933 | 0.962 |
| | AF | 0.977 | 0.875 | 1.000 | 0.972 | 1.000 | 0.988 | 0.933 | 0.938 |
| | ST | 0.953 | 0.778 | 1.000 | 0.944 | 1.000 | 0.960 | 0.875 | 0.761 |
| | Weighted mean | 0.942 | 0.781 | 0.981 | 0.949 | 0.905 | 0.932 | 0.833 | 0.799 |

*1dAVB* 1st degree AV block, *AF* atrial fibrillation, *AUPRC* area under precision recall curve, *AUROC* area under receiver operator characteristic curve, *LBBB* left bundle branch block, *NPV* negative predictive value. *PPV* positive predictive value, *RBBB* right bundle branch block, *SB* sinus bradycardia, *ST* sinus tachycardia.

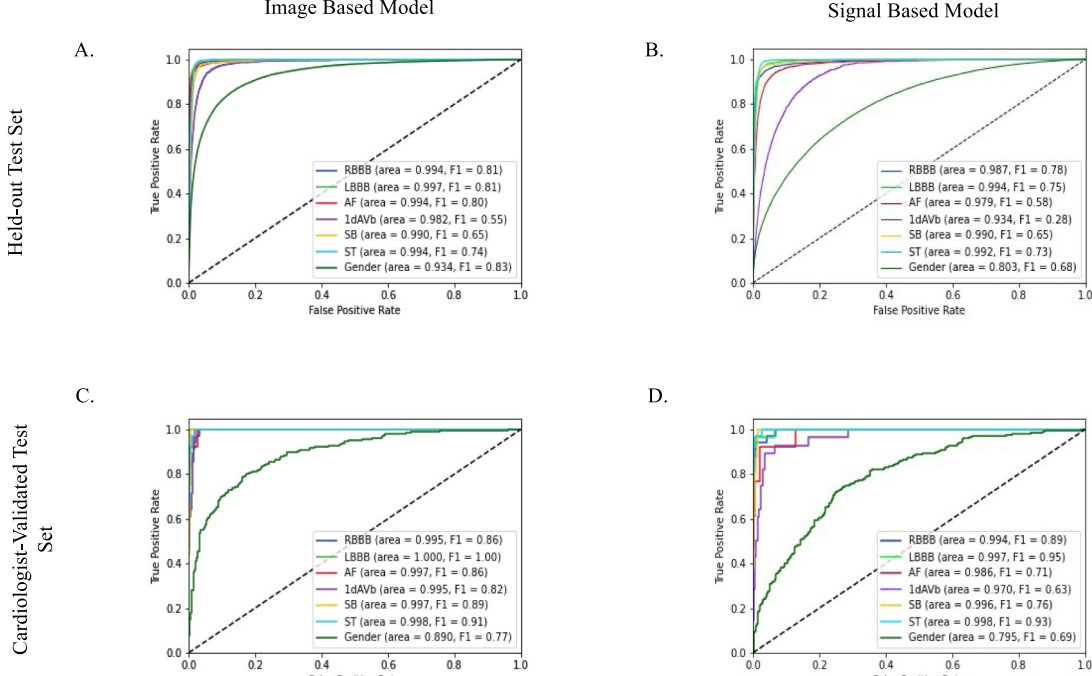

**Fig. 2 Area under the receiver operator characteristic (AUROC) curves for image and signal-based models.** AUROC on held-out test set (**A**, image model; **B**, signal model) and cardiologist validated test set (**C**, image model; **D**, signal model) are included in the 4 image panels. 1dAVB 1st degree AV block, AF atrial fibrillation, LBBB left bundle branch block, RBBB right bundle branch block, SB sinus bradycardia, ST sinus tachycardia.

Our study has certain limitations. First, while our model has excellent performance characteristics, the reason for the discordance of the model and the labels could not be confirmed for all ECGs. The ECGs have been reviewed by a cardiologist in our training data[3], and by two cardiologists in the external validation data[13], but we found that high probability predictions initially noted to be false positives in both these sets actually represented inaccurate labels. Moreover, this pattern was not observed in the cardiologist-validated internal test set, further suggesting that the performance of the model likely exceeds what is reported in the held-out and external validation sets, which could not be independently validated given the large size of the data.

Second, we focused on 6 clinical labels, based on their availability in the training data, and therefore, our models would not

apply to other clinical disorders. We believe that our study identifies a strategy of leveraging ECG images for a broad set of disparate diagnoses—spanning rhythm and conduction disorders, as well as hidden labels. We also demonstrate how existing data repositories with waveform data can be augmented to accomplish this task. Our goal for future investigations will be to design custom models on repositories with a broader set of labels as well as extract waveform-specific measures through access to valid information from richer data repositories.

Finally, we were unable to discern an interpretable pattern from Grad-CAM on ECG-based identification for gender classification, but the performance of the model on external validation data argues against overfitting. Moreover, Grad-CAM does not allow for interpretability of rhythm disorders, where large

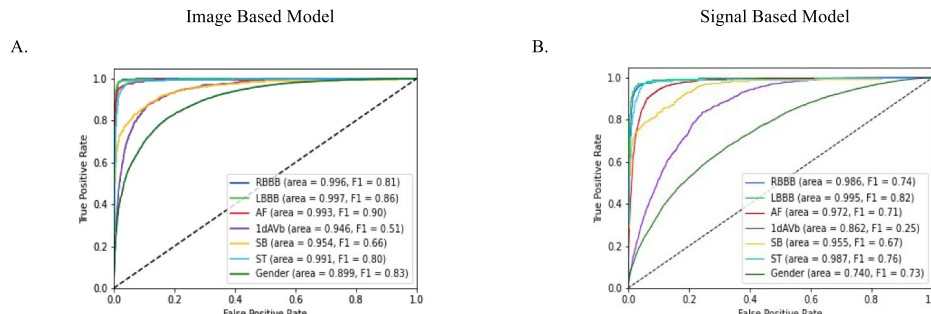

**Fig. 3 Area under the receiver operator characteristic (AUROC) curves on the external validation set (PTB-XL).** **A** Image based model, and **B** Signal based model. 1dAVB 1st degree AV block, AF atrial fibrillation, LBBB left bundle branch block, RBBB right bundle branch block, SB sinus bradycardia, ST sinus tachycardia.

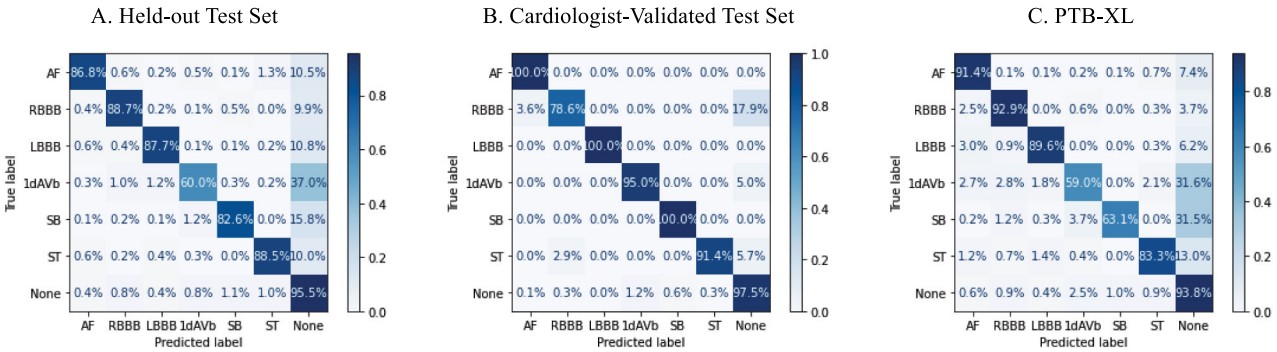

**Fig. 4 Confusion matrices for image model predictions.** **A** Held-out test set, **B** cardiologist-validated test set, and **C** PTB-XL. 1dAVB 1st degree AV block, AF atrial fibrillation, LBBB left bundle branch block, *RBBB* right bundle branch block, SB sinus bradycardia, ST sinus tachycardia.

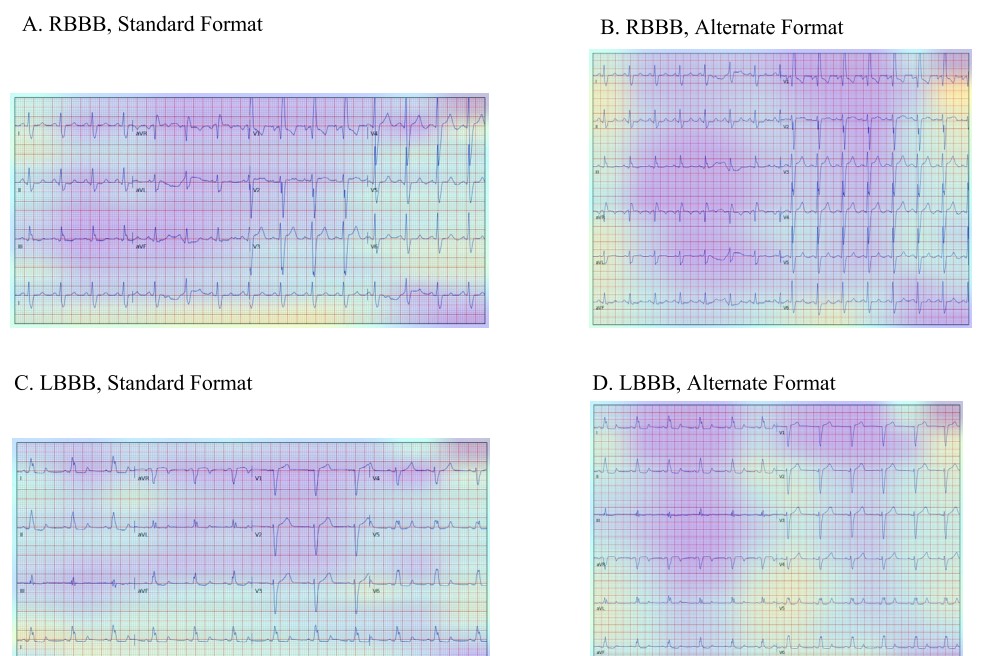

**Fig. 5 Gradient-weighted Class Activation Maps for Image Predictions.** Average of the Gradient weighted Class Activation Maps (Grad-CAMs) for the 25 most confident predictions of right bundle branch block (RBBB) and left bundle branch block (LBBB) on standard and alternate format images. **A** RBBB in standard format, **B** RBBB in alternate format, **C** LBBB in standard format, and **D** LBBB in alternate format.

sections of repeating patterns in the ECG, which cannot be defined by heatmaps, are required for diagnosis. While the model performed well on these labels, they represent a challenge for the interpretability of AI applications on ECG tracings.

In summary, we have developed an externally validated multilabel automated diagnosis algorithm that accurately identifies rhythm and conduction disorders from either ECG images or raw electrocardiographic signals. The versatility, interpretability, generalizability,

and broad ability to incorporate ECG images can broaden the application of AI to clinical electrocardiography.

## Methods

The study was reviewed by the Yale Institutional Review Board, which waived informed consent as the study uses deidentified data.

**Data source and study population**. We used 12-lead ECGs collected by the TNMG and described previously in Ribeiro et al.[3]. The data were assembled as a part of the CODE study[6]. They include deidentified signal data captured between 2010 and 2017 from 811 out of the 853 municipalities in the state of Minas Gerais, Brazil. The ECGs obtained were collected using one of two models of tele-electrocardiograph machines, the TEB ECGPC, manufactured by Tecnologia Electronica Brasileira, or the ErgoPC 13, manufactured by Micromed Biotecnologia. These ECGs were recorded as standard 12-lead recordings sampled at frequencies ranging from 300 to 600 Hz for 7–10 s. In addition, information on patient demographics and six clinical labels were available.

Briefly, labels for the primary CODE study dataset were obtained through the following procedure, as described by Ribeiro et al.[3,6]. Automated University of Glasgow statements and Minnesota codes obtained by the Uni-G automatic analysis software were compared to both automatic measurements provided by the Uni-G software and text labels extracted from expert reports written upon initial reading of the signals. These labels were extracted using a semi-supervised Lazy Associative Classifier trained on a dictionary created from text reports. Discrepancies in the labels provided by extracted expert annotation and automatic analysis were settled using both cutoffs related to ST, SB, and 1dAVb, as well as manual review.

Further, to augment the evaluation of models built on the primary CODE study dataset, where clinical annotations were derived from routine clinical care, a secondary cardiologist-validated annotation test dataset was used. This consisted of additional ECGs obtained from the TNMG network between April and September 2018. These ECGs were rigorously validated by 2-to-3 independent cardiologists based on criteria from the American Heart Association[7]. This represents the cardiologist-validated test set from the study by Ribiero et al[3]. Finally, an ECG dataset from Germany was obtained for external validation (described in the section on external validation).

**Data preprocessing**. We resampled all ECGs to a 300 Hz sampling rate. Such a down-sampling of signals represents a standard preprocessing step for ECG waveform analyses to allow for the standard data structure required for modeling. QRS peak detection by the XQRS algorithm in lead I of the original recording aided in the determination of the start and end of the recordings to be assessed with the signal models and to be converted to ECG images[8]. ECGs that did not have any peaks detected (94,277, or 4.1%) were discarded, as inspection of a sample of 50 ECGs after further preprocessing did not have any cardiac activity but rather uninterpretable noise.

For signal data, 5 s simultaneous recordings across leads were used to obtain uniform length inputs. Each ECG was processed as a matrix of 12 leads of 1500 sequential data points, representing a 5 s acquisition at a 300 Hz sampling frequency. Each sample started 300 ms before the second QRS peak detected by the XQRS algorithm in lead I of the original recording. The 5 s samples were preprocessed with a one-second median filter applied across each lead and subtracted from the original waveform to remove baseline drift. Next, we employed a peak annotation technique for each ECG to include ECG morphology along with waveform data as an input to the signal model. For all ECGs, we found the median and standard deviation of all RR, PR, QRS, and ST intervals for each lead using NeuroKit2, a Python toolkit for signal processing[21]. Briefly, the NeuroKit2 algorithm is based on the detection and delineation algorithms of Martinez et al.[22], and uses a discrete wavelet transform to localize QRS peaks, allowing it to identify the local maxima associated with these peaks regardless of noise artifacts that may be present in the signal. It then performs a guided search for P and T waves based on the information about QRS location and known morphologies for these waves in electrocardiographic signals. The median lengths of each interval across leads were included as additional inputs into the signal model (Fig. 1).

The ECGs were transformed to the corresponding printed images using the python library ecg-plot[9]. From the training waveforms, two image-based datasets were created for image model training, including a standardized subset, and a real-world variation subset. In the standardized subset, each pre-image transformation sample started 300 ms before the second QRS peak detected in lead I. To ensure that like a clinician the model learned about lead-specific information based on the label of the lead, rather than just the location of leads on a single ECG format, we included four formats of images in the ideal dataset (Fig. 1). Standard printed ECGs in the United States typically consist of four 2.5 s columns presented sequentially on the page, with each column containing three leads of 2.5 s intervals of a continuous 10 s record. A 10-s rhythm strip was generated using a concatenation of lead I signal on the last and first QRS of the same 5 s signal[7]. The second plotting scheme we chose consisted of two columns each containing six 5-second recordings, with one column containing simultaneous limb leads and the other

simultaneous precordial leads. We treated this scheme as our alternate image form (Fig. 1).

The third and fourth formats were representations of the standard and alternate formats with shuffled lead locations. In the shuffled standard format precordial leads were presented in the first two columns and limb leads in the third and fourth. In shuffled alternative images precordial leads were presented in the first column and limb leads in the second (Fig. 1). This was done to prevent overfitting to certain locations on the ECG image and to create a more versatile algorithm.

The second real-world variation subset included images based on the same waveform signals as the standardized subset. While in the standard format of the standardized subset, the variations expected in real-world ECG images were introduced in constructing these additional images. These variations fell into two categories. First, each pre-image transformation sample started at a point that was selected from a uniform distribution of 0–1.3 s offset from the first detected QRS peak in lead I. This was designed to mimic the lack of predetermined starting points in real-world ECGs. The second variation introduced in the real-world variation dataset was different color schemes for the background of the ECG image and the signal being plotted. We modified the ecg-plot software to produce ECGs with three common color schemes to the ECGs in the ideal dataset. This included two variations of black and white ECGs, as well as a format with a pink background but a black signal (Fig. 1).

**Study outcomes**. Each ECG was annotated for six physician-defined clinical labels spanning rhythm and conduction disorders (AF, RBBB, LBBB, 1dAVB, ST, and SB). Diagnoses for ECGs were obtained by a combination of the automated diagnosis provided by the Glasgow ECG analysis software and the natural language processing (NLP) extracted diagnosis from a cardiologist report of the ECGs. To detect whether our image and signal-based modeling approaches could detect hidden or higher-order features that are not discernable from the ECG by human readers, we defined patient sex as the seventh label given its consistent availability across the study population and prior literature describing its detection from signal-based electrocardiographic deep learning models[16].

**Experimental design**. All ECGs were randomly subset into training, validation, and held-out test sets (90%-5%-5%). Given the low prevalence of all clinical labels, the data splits were stratified by clinical labels, so cases of each of the six labeled clinical disorders were split proportionally among the sets, as was gender. To ensure that model learning was not affected by the low frequency of certain labels, custom loss functions based on the effective number of samples class sampling scheme were used for both image and signal models, with weighting based on the number of samples for each class (Supplementary Table 2)[11]. To account for class imbalance gave higher weights to rarer classes with the goal of ensuring that performance on metrics sensitive to class imbalances remained high.

**Image model overview**. For image data, we built a CNN model based on the EfficientNet set of architectures[10]. To balance the complexity and accuracy of a model with the computational resources required for training, we chose the B3 architecture, which required PNG images to be sampled at $300 \times 300$ square pixels. The model includes 384 layers, has over 10 million trainable parameters, and is composed of seven blocks (Supplementary Fig. 1). Additional dropout layers were added to prevent overfitting, and the output of the EfficientNet-B3 model was fed into a fully connected layer with sigmoid activation to predict the probability of target labels. The model was trained with an Adagrad optimizer and learning rate $5 \times 10^{-3}$ for seven epochs with a minibatch size of 64. The optimizer and learning rate was chosen after hyperparameter optimization. We a priori chose 30 epochs for training models with built in early stopping based on validation loss not improving after three consecutive epochs. Model weights were initialized as the pretrained EfficientNetB3 weights on the ImageNet dataset to take advantage of any possible transfer learning.

We trained our primary image model on both the standardized images, with a 40%-40%-10%-10% split of standard, alternate, standard shuffled, and alternate shuffled images as well as the real-world variation subset, with equal numbers of the three background colors stratified in the same manner described above (see schematic in Fig. 1). Validation was conducted on standardized images. All images were rotated a random amount between −10 and 10 degrees prior to being input into the model for training and validation to mimic variation that might be seen in uploaded ECGs and further aid in the prevention of overfitting.

**Signal model overview**. For signal data, we developed a custom CNN model that combined inception blocks, convolutional, and fully connected layers, similar to Raghunath et al.[4]. Our model had two inputs, the $1500 \times 12$ waveform data, and the $8 \times 1$ array of derived elements from the ECG including various standard intervals.

The waveform data input was passed through seven branches, each one containing an initial 1D-convolutional layer followed by 5 inception blocks and another 1D-convolutional layer (Supplementary Fig. 2). We used inception blocks to capture information of various kernel sizes and therefore differing assumptions of the locality, or spatial connectivity of ECG data. For example, smaller kernels

performed more localized learning, on individual waves or parts of waves, while larger ones combined data across sections of multiple waves.

The signal in each of these branches was then flattened, passed through two fully connected layers, and concatenated with data from the other branches. These final waveform signal data were passed through a fully connected layer and then concatenated with the 8 × 1 peak morphology input. This information was passed through two more fully connected layers, the second of which had sigmoid activation to predict the probability of target labels. Further details of the signal models are included in Supplementary Methods in the Online Supplement.

In sensitivity analyses, we trained a signal model with the exact same architecture as the one described, but without peak morphology inputs to test the ability of a convolutional network to perform these operations internally and learn the same patterns.

**Internal validation**. Model performance was evaluated on the 5% held-out test set. In addition to the held-out test set, we validated the algorithm in an expert validated test of 827 ECGs from distinct patients in Minas Gerais, Brazil during a different time window (April to September 2018). The labels on these ECGs were validated by 2 cardiologists, with disagreements resolved by a third cardiologist, based on AHA criteria. For the image-based models, the performance was evaluated separately for the both the standard and alternate ECG lead layouts of the held-out and the cardiologist validated test sets.

**External validation: ECG waveform database**. We also pursued external validation to assess the ability to generalize to novel data sources[23]. In addition to the held-out test set, model performance was evaluated on the Germany-based external validation dataset, PTB-XL, whose data have been previously described[13]. Briefly, the dataset has 21,837 recordings from 18,885 patients. ECGs were collected with devices from Schiller AG between 1989 and 1996 and are available as 10 s recordings sampled at 500 Hz for 10 s. Each record has labels for diagnostic, form, and rhythm statements. Data were transformed in the same manner as the data from Brazil, and the same labels were extracted to assess model performance.

**External validation: real-world ECG images**. We pursued validation on two real-world image datasets. The first of these was from the Lake Regional Hospital System (LRH), a rural US hospital in Osage Beach, MO. These data included 64 ECG images including 8–10 ECGs for our six labels of interest, as well as ECGs that were labeled as normal. Subjectively, the ECGs had a similar layout as the standard ECG format but had the V1 lead rather than lead I as the rhythm data. There were vertical lines demarcating the leads, the signal was black rather than blue, and the background color and grid of the ECGs varied, as did the location and the font of the lead label.

The second real-world image dataset consisted of ECG images available on the internet, representing 42 ECGs. We followed a systematic approach to constructing this dataset. We first accessed images on life in the fast lane (LIFTL) website, an educational website for teaching about ECGs available at https://litfl.com/ecg-library/. From LIFTL, we took all ECGs for the labels of interest, without any preselection. Our goal was to have at least 6–7 ECGs per label. As LIFTL only had 2–6 ECGs per label, we pursued a google image search for "<label> ecg image" and selected the first 12-lead ECGs that appeared until we had seven ECGs for each label of interest. Qualitatively, these web ECGs included both standard and alternate format images, as well as a new format of images such as ones without rhythm leads, or with two or more rhythm leads. Moreover, the ECGs had varying background colors, signal colors, quality, and location of lead labels, and many had additional artifacts that were not present in our training data. These images are available from the authors upon request.

**Model interpretability**. We used Gradient-weighted Class Activation Mapping (Grad-CAM) to highlight the regions in an image predicting a given label[14]. This provides interpretability of the model's predictions and allows for the evaluation of whether model-assigned labels are based on clinically relevant information or on heuristics based on spurious data features[15]. To deploy Grad-CAM in our models, we calculated the gradients of the final stack of filters in the network for each prediction class of interest. These gradients assigned the importance of a given pixel to the prediction of the label. Then, we created filter importance weights by performing a global average pooling of the gradients in each filter, emphasizing filters whose gradient suggested they contributed to the prediction of the class of interest. Finally, each filter in our final convolutional layer was multiplied by its importance weight and combined across filters to generate a Grad-CAM heatmap. We overlayed the heatmaps on the original ECG images.

We used two approaches to assess model interpretability. First, we examined individual ECGs, obtaining, and overlaying the Grad-CAM heatmaps for both the standard and alternate images. Second, we averaged class activation maps for a condition of interest for any given model. We used a simple mean across the heatmaps for the images for a certain condition, and overlayed this average heatmap over a representative ECG to understand it in context.

We chose LBBB and RBBB to illustrate the interpretability as these labels have lead-specific information that is used to make the clinical diagnosis. This contrasts with the other labels (AF, ST, SB, 1dAVB) where the information on these rhythm disorders can be deduced from any of the ECG leads, limiting assessment of lead-specific learning.

**Statistical analysis**. Model performance was evaluated in the held-out test set, cardiologist-validated test set, and the external validation set. We assessed the area under the receiver operating characteristic (AUROC) curve, which represents model discrimination, with values ranging from 0.50 to 1.00, representing random classification and perfect discrimination of labels, respectively. In addition, we assessed the area under the precision recall curve (AUPRC), and F1 score of the model for each label, metrics that are sensitive to rare events and may provide more insight on the clinical usefulness of our models[23–25]. We also assessed the sensitivity, specificity, positive predictive value, and negative predictive value for each label. For the threshold dependent measures, the threshold that maximized the F1 score was selected, representing a strategy that optimized both precision and recall. Class weighted mean metrics across clinical labels were calculated to evaluate the performance of a model at a dataset-level by taking the weighted average of metrics on the labels in that dataset (e.g., held-out test set, cardiologist-validated internal test set, external validation set), weighed by the counts of labels in that dataset. Models were compared by computing $p$ values and 0.95 confidence intervals for AUROC using DeLong's Method[26]. We used confusion matrices to illustrate the discordance between a model's predictions and the diagnoses that came with our datasets. These were constructed among ECGs with single clinical labels. All analyses were performed using Python 3.9.

**Reporting summary**. Further information on research design is available in the Nature Research Reporting Summary linked to this article.

## Data availability

The cardiologist-validated test set is publicly available on the link: https://zenodo.org/record/3765780#.YVIM8J5Kgl9%2F. The training data are based on the CODE study, published in Nature Communications (Volume 11, Article number: 1760 (2020)) and can be obtained from Ribeiro et al. (antonio.ribeiro@ebserh.gov.br). Restrictions apply to the availability of the training set and requests to access will need to be submitted and reviewed on an individual basis by the Telehealth Network of Minas Gerais for academic use only. The data for PTB-XL are available from physionet.org. The test data from Lake Regional Health System and the web-based validation set will be made available from the authors upon request.

## Code availability

The code for the study was shared with the Editors and reviewers for peer review and is available from the authors upon request for replicating the results. The algorithm developed in this study is the intellectual property of Yale University. The model for testing electrocardiographic images is available as a publicly accessible web-based tool at https://www.cards-lab.org/ecgdx.

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

## Acknowledgements

This study was supported by research funding awarded to Dr. Khera by the Yale School of Medicine and grant support from the National Heart, Lung, and Blood Institute of the National Institutes of Health under the award K23HL153775. The funders had no role in the design and conduct of the study; collection, management, analysis, and interpretation of the data; preparation, review, or approval of the manuscript; and decision to submit the manuscript for publication.

## Author contributions

V.S. and R.K. conceived the study and drafted a research plan. R.K. accessed the data in collaboration with A.L.P.R. and A.H.R. V.S. and R.K. conducted the analyses and drafted the manuscript. All authors provided feedback regarding the study design and made critical contributions to the writing of the manuscript. R.K. supervised the study, procured funding, and is the guarantor.

## Competing interests

B.J.M. reported receiving grants from the National Institute of Biomedical Imaging and Bioengineering, National Heart, Lung, and Blood Institute, US Food and Drug Administration, and the US Department of Defense Advanced Research Projects Agency outside the submitted work; in addition, B.J.M. has a pending patent on predictive models using electronic health records (US20180315507A1). A.H.R. is funded by *Kjell och Märta Beijer Foundation*. D.L.J. has received personal fees from MyoKardia and has received a grant through the SHaRe Cardiomyopathy Registry, which is funded by MyoKardia. W.L.S. was an investigator for a research agreement, through Yale University, from the Shenzhen Center for Health Information for work to advance intelligent disease prevention and health promotion; collaborates with the National Center for Cardiovascular Diseases in Beijing; is a technical consultant to Hugo Health, a personal health information platform, and co-founder of Refactor Health, an AI-augmented data management platform for healthcare; is a consultant for Interpace Diagnostics Group, a molecular diagnostics company. H.M.K. works under contract with the Centers for Medicare & Medicaid Services to support quality measurement programs, was a recipient of a research grant from Johnson & Johnson, through Yale University, to support clinical trial data sharing; was a recipient of a research agreement, through Yale University, from the Shenzhen Center for Health Information for work to advance intelligent disease prevention and health promotion; collaborates with the National Center for Cardiovascular Diseases in Beijing; receives payment from the Arnold & Porter Law Firm for work related to the Sanofi clopidogrel litigation, from the Martin Baughman Law Firm for work related to the Cook Celect IVC filter litigation, and from the Siegfried and Jensen Law Firm for work related to Vioxx litigation; chairs a Cardiac Scientific Advisory Board for UnitedHealth; is a member of the IBM Watson Health Life Sciences Board; is a member of the Advisory Board for Element Science, the Advisory Board for Facebook, and the Physician Advisory Board for Aetna; and is the co-founder of Hugo Health, a personal health information platform, and co-founder of Refactor Health, a healthcare AI-augmented data management company. A.H.R. is supported in part by CNPq (310679/2016-8 and 465518/2014-1) and by FAPEMIG (PPM-00428-17 and RED-00081-16). R.K. is the coinventor of U.S. Provisional Patent Application No. 63/177,117, "Methods for neighborhood phenomapping for clinical trials" and a co-founder of Evidence2Health, a precision health platform to improve evidence-based care. The other authors have no relevant competing interests.
