## [Peer Review File · Nature Communications]

Reviewers' Comments:

Reviewer #1:

Remarks to the Author:

This paper proposed an AI-based solution for multilabel diagnosis of ECG 2-D images and 1-D signals. The reviewer has the following concerns:

1. Developing a model on printed ECG images rather than ECG signals addresses an important need in ECG automatic diagnosis, since signal-based method has difficulty in real-time and explainable applications, especially in remote environment. So it is important to explore the image-based model. An essential concern is that, image-based model is heavily dependent on the used ECG-image definition format. The authors should include the commonly used ECG-image formats. The reviewer suggests to include the introduction about the ECG-image standard and definition from different countries or representative device providers. The current version only includes two image format and their shuffled versions.
2. Another concern is that, the authors involved a large amount of ECG data, however, the number of classification for ECG abnormalities was small (only 6). Thus, the practical value of the developed models is limited. As comparison, the PhysioNet/Computing in Cardiology challenge 2020 involved 121 types of ECG abnormalities and challenged the classification task of 27 types (see <https://physionetchallenges.org/2020/>). The current classification task is relatively simple. The reviewer suggests to add more ECG abnormalities to improve the practical value of the model.
3. Further concern is that, how to validate the accuracies of ECG labels for all 2,228,236 12-lead ECGs? This question should be clearly clarified.
4. Can the image-based model extract the ECG information like the channel (I, II, III, V1, etc.)? or ECG waveform features like QRS amplitude, hear rate? These explainable features outputs can enhance the model performance on multi-label classification tasks or more complicated tasks.

Reviewer #2:

Remarks to the Author:

This is a high quality and interesting study by Sangha and colleagues focusing on the application of deep neural networks to ECG data to perform labeling of selected clinical arrhythmia and conduction disorder patterns. This is an area that has seen considerable attention in recent years, notably the studies from Hannun et al. (single lead digital signal, ambulatory ECG), Ribiero et al (12-lead digital signal), and Gliner et al (12-lead digital signal, plus image-based model for atrial fibrillation), and this work extends these works in important ways. In particular, the potential of an image-based neural network for ECG rhythm detection is a primary focus of this work because it is able to process data in settings where access to the digital signals is not readily available. The authors present analyses within the large Clinical Outcomes in Digital Electrocardiology (CODE) dataset from Brazil, as well as an external dataset from Germany. As expected (based on the works Hannun and Ribiero), the models demonstrate excellent performance in the multi-label classification task, including excellent performance in the independent test set and robustness to the specific format of the ECG image (which I thought was a particularly clever design element). What is surprising, however, is that not only was the image-based neural network comparable to the signal-based model (reported in the Gliner study), its performance appears to be generally superior. Moreover, the inclusion of Grad-CAM results helps to demonstrate that the models are cueing on clinically relevant features for classification (where human readers have an understanding of what cues to look for). Overall, I thought this was strong work, but do have some questions and suggestions for potential consideration and improvement.

1. There are several pre-processing steps described in the generation of the ECG images; for example, resampling to 300 Hz and starting the signal readout at a pre-determined time with respect to the QRS peak. I think clarifying the rationale for these steps is important (eg, are those considered standard pre-processing steps for ECG printing?) because otherwise, such manipulation runs counter to the premise of the work in seeking to make the compatibility of ECG data robust for neural network ingestion. In other words, are these images more idealized and regularized than what may be encountered in 'the real world'?
2. Similarly, since the study is focused on characterizing the performance of an image-based ECG deep learning model, I think it would be appropriate to add an additional sensitivity analysis with respect to varying image quality and/or noise characteristics, particularly given that scanned

images are stated use case.

3. It was a rather surprising finding to me that the image-based model outperformed the signal-based model, especially considering that the image-based model theoretically does not contain any extra information and the signal-based model even had additional inputs (median intervals, etc). Despite this surprising finding, there was not much consideration or speculation as to the source of these findings in the discussion. I think some additional discussion on this point is warranted.

Some additional minor points:

- I assume that the 827 ECGs in the Cardiologist-Validated test set matches the same 827 ECG test set from the Ribiero study but, if so, this point should be made more explicitly. Also, if so, is the cardiologist reader from the Ribiero study one of the 2-to-3 cardiologist readers for this analysis? Finally on this point – did the cardiologist reviewers have access to the prior clinical interpretations, or were their reviews blinded to any other interpretation?
- With respect to the generation of ECG images with varying formats – were studies duplicated across formats within the held-out test set, or was each study only represented once?
- In the paragraph describing the results for the held-out test set for the image data, I believe there is a typo in describing the specificity at maximum F1 (0.88 instead of, presumably, 0.98).
- With respect to the confusion matrices in Figure 4, it is unclear how the reported 'accuracy' numbers are readily derived from the numbers presented in the figure and why they differ so drastically from those reported for the same labels in Table 1.
- There are no details provided for the peak annotation technique mentioned in the 'Data Preprocessing' section of the Methods.

Reviewer #3:

Remarks to the Author:

The paper describes applying two different types of convolutional neural networks (CNNs) on ECG images transformed from ECG signals and raw ECG signals respectively to identify 6 clinical labels as well as the subject gender. The results demonstrate that ECG images, similar as signals, can be used to classify the 6 clinical disorders and define hidden features for identifying the subject gender. Also, it shows the application of Gradient-weighted Class Activation Mapping (Grad-CAM) to highlight the regions in the image predicting a given label which provides interpretability.

The work is creative in a way that it implements an automated diagnosis model (with a deep neural network) for ECG images in addition to ECG signals. As the author mentioned in the paper, ECGs are frequently printed and scanned as images. However, the ECG images used in the study are generated from the ECG signals directly using the `ecg-plot` python library. The plotted ECG images are very different than the printed and scanned ECG images for the color, light, shadow and integrity sometimes, which makes the study less practical.

The study is based on a dataset including 2,228,236 12-lead ECGs from 1,506,112 patients, which makes the results convincing. Moreover, the evaluation of the model also includes a secondary cardiologist-validated annotation test dataset that can assess the model trained from dataset that annotated from a single clinician. In addition, an external validation set collected from another country avoids the doubts about bias caused by the population distribution. The number of ECGs with at least one of the six disorders (231,704) is a small amount compared to the whole dataset, which may affect the model since it sees much more normal cases, and the effect is better to be explored and discussed. Similarly, the ECGs with the six clinical labels (AF, RBBB, LBBB, 1dAVb, SB and ST) are skewed with more RBBB (61,551) and SB (48,296) than others (from 34,446 to 39,661), while there is no discussion about the effects of the skewness in data.

The paper brings some interesting points for processing the inputs such as using different leads layout in the image to ensure that "the model learned about lead-specific information based on the label of the lead, rather than just the location of leads on a single ECG format". It also applies the Grad-CAM method to identify the ECG regions that are most important for the label classification. However, it does not have enough analysis on whether or not using the different image formats achieve the goal of preventing the model from just learning the location of leads on a single ECG format.

The paper emphasizes the development of "dual modality multilabel prediction algorithms that incorporate either ECG images or signals as inputs." However, it does not provide any integration mechanism to coordinate the image classification model which is based on an Efficientnet-B3 architecture, and the signal classification model which is based on an inception architecture. In addition, the paper does not include studies considering the characteristics about ECG images and/or ECG signals, and lacks analysis on model selections and comparisons.

Overall, the study shows promising results on using an Efficientnet-B3 network to classify 6 clinical disorders as well as genders from ECG images generated from ECG signals. It is based on a valuable large ECG dataset as well as distinct and external evaluation sets to show the generalization of the model. However, because the plotted ECG images are different than the printed ones, it reduces the ability of using the model for the real-world applications. Also, it lacks analysis on the skewness in data and the effects of different layouts for the input image. Moreover, it does not show any integration mechanism for the two models of the dual modality multilabel prediction system.

RESPONSE TO REVIEWER COMMENTS

(Sangha et al. 2021)

EDITORS' COMMENTS

Comment 1: We generally agree with Reviewer #3 that a more thorough analysis of the effect of real-world noise effects on image acquisition on model performance is needed to demonstrate clinical application more convincingly.

Response: We thank the Editors and reviewers for their careful consideration of our work and for their constructive feedback. We have provided a point-by-point response to their comments below.

We recognize the importance of demonstrating the robustness of models to noise, which is commonly seen in real-world clinical application. The primary purpose of the model reported in our original submission was to demonstrate the effectiveness of an ECG model that was trained on either images or signals in both detecting clinical labels as well as a hidden label. We had focused on generalizability and interpretability of such an approach.

However, we appreciate that the image-based model would need to be robust to variability in real-world data, as the intended application would require real-world ECG images rather than those generated from signals. This variability was not a part of the original training, where we only varied the training sample on lead layout. We explicitly confirmed this by testing the model on a subset of 5000 ECGs in our held-out test set, wherein we introduced changes in background color, start point of ECG signals, and rotations, and found a lower performance (Table R1).

Table R1: Class Weighted Mean Performances of Original Model on Subsets of Held-Out Test Set with Noise Effects Introduced

Dataset	Accuracy	PPV	NPV	Specificity	Sensitivity	AUROC	F1	AUPRC
Standardized Subset	0.987	0.656	0.996	0.991	0.82	0.99	0.727	0.707
Black and White	0.979	0.476	0.991	0.986	0.568	0.946	0.511	0.439
Rotated	0.981	0.539	0.992	0.989	0.616	0.968	0.572	0.534
Shifted Starting Point	0.987	0.646	0.996	0.991	0.775	0.986	0.696	0.663

However, we posited that these inputs can be explicitly factored into model development. The study team included 4 cardiologists with broad experience in electrocardiography. Based on our experience with real-world ECGs, we hypothesized that real-world variability – beyond lead layout - would be represented in many distinct categories, but that 3 of them would be implementable in training and capture a majority of the differences: **1)** differences in the background color and thickness and color of grid lines on printed ECG images **2)** differences in the angle and size of uploaded ECG images, and **3)** variations in the starting point of the actual ECG recording, as our original training data was selected as a 5 second sample starting 300 ms before the 2nd detected QRS complex.

To develop this real-world adapted model, in addition to training on the original training set (now called the standardized subset) that included 2,005,412 ECGs across 4 formats, we augmented the sample with an additional 2,005,412 ECG images that represented variations in the background color and starting point for the original training data set. These additional ECGs had a starting point that was selected from a uniform distribution of 0 to 1.3 second offset from the first detected QRS complex in the signal. The background color and color of the signal trace was modified to include three different color variations (**Figure R1, Figure 1 in the manuscript**). All of these transformations were stratified by clinical labels. Additionally, all 4 million images underwent a random rotation between -10 degrees and +10 degrees during the model training process. We have now revised the manuscript to describe these two training subsets. We have included **Figure 1** that provides a schematic of the training of the real-world Model and described this process Image-based Classification section of the Results (page 5 paragraph 4).

“From the training waveforms two image-based datasets were created for image model training, including a standardized subset, and a real-world variation subset.”

(Results, page 6 paragraph 3)

“The second real-world variation subset of the training sample included images based on the same waveform signals as the standardized subset. All ECGs were in the standard layout of the standardized subset, but variations expected in real-world ECG images were introduced in constructing these additional images. These variations fell into two categories. First, each pre-image transformation sample started at a point that was selected from a uniform distribution of 0 to 1.3 second offset from the first detected QRS peak in lead I. This was designed to mimic the lack of predetermined starting points in real-world ECGs. Second, color schemes for the background of the ECG image plotted from the signals were varied across ECGs (Details in Methods). (Figure 1).”

(Results, page 7, paragraph 2)

“All images were randomly rotated between -10 and +10 degrees prior to being input into the model for training and validation to mimic variation that might be seen in uploaded ECGs and further aid in prevention of overfitting.”

We additionally included a description of these training set augmentations in the Methods (page 22, paragraph 2)

“The second variation introduced in the real-world variation dataset was different color schemes for the background of the ECG image and the signal being plotted. We modified the ecg-plot software to produce ECGs with three common color schemes to the ECGs in the ideal dataset. This included two variations of black and white ECGs, as well as a format with a pink background but black signal (Figure 1).”

Figure 1: Study Outline

This new model trained on noise-related artifacts anticipated in real world data was more robust in identifying the labels in noisy held-out test sample (the same as Table R1 above). Specifically, the updated model did not demonstrate any drop-offs in performance as various types of noise were added (**Table R2, Supplementary Table 3**). While this was expected, we explicitly confirmed it in our assessment. Moreover, this updated model was tested in other external validation sets as well as real-world ECGs, which is discussed in further detail below.

Table R2 / Supplementary Table 3: Class Weighted Mean Performances of Real-World Model on Subsets of Held-Out Test Set with Noise Effects Introduced

Dataset	Accuracy	PPV	NPV	Specificity	Sensitivity	AUROC	F1	AUPRC
Standardized Subset	0.989	0.688	0.997	0.992	0.841	0.992	0.755	0.736
Black and White	0.989	0.687	0.997	0.992	0.847	0.992	0.756	0.732
Rotated	0.989	0.687	0.997	0.992	0.847	0.992	0.757	0.734
Shifted Starting Point	0.989	0.704	0.996	0.993	0.816	0.992	0.753	0.743

We described the process of testing the robustness of our real-world model to artificial noise in the Results section (page 8, paragraph 3).

“To explicitly test the robustness of this model to the noise-artifacts we anticipated in the real world, we tested the model on a representative subset of 5,000 ECGs in our held-out test set. We compared model performance on four variations of the same 5,000 ECG waveform data. They were plotted in the standard ECG layout, first without additional variation, second, in black and white, third, with random rotations between -10 and +10 degrees, and fourth, with a varying starting point for plotted ECG with a 0 to 1.3 second offset from the first detected QRS complex in the signal. We saw no drop off in performance of the model as various types of noise were added (Supplementary Table 3).”

We also tested this real-world adapted model on PTB-XL, our external validation dataset, without any noise. The updated model trained on noisy data outperformed our original model across labels. We speculate that this may represent the model learning novel information through more variation in the training set. We have presented the performance of the new real-world model on PTB-XL in **Table R3** below and have included this data in the updated **Table 2**.

Table R3: Performance of Image Model on PTB-XL

Label	Accuracy	PPV	NPV	Specificity	Sensitivity	AUROC	F1	AUPRC
Male	0.81	0.784	0.845	0.738	0.876	0.899	0.827	0.904
1dAVb	0.96	0.457	0.984	0.974	0.573	0.946	0.508	0.457
RBBB	0.989	0.712	0.998	0.99	0.939	0.996	0.81	0.793
LBBB	0.993	0.823	0.998	0.995	0.898	0.997	0.859	0.902
SB	0.98	0.673	0.989	0.991	0.641	0.954	0.656	0.641
AF	0.986	0.898	0.992	0.992	0.898	0.993	0.898	0.934
ST	0.984	0.759	0.994	0.99	0.838	0.991	0.796	0.809
Weighted Mean	0.982	0.743	0.992	0.989	0.805	0.981	0.77	0.776

We recognize, however, that the held-out test and external validation may be insufficiently varied, because real-world printed ECGs would include other artifacts that we may not have anticipated. However, we posited that the model trained on data with variation on layouts, colors, rotations, as well as ECG start points, will be better positioned to learn underlying information in noisy data. We evaluated these assumptions in 2 real-world ECG datasets, that we have elaborated on in our response to Comment #2. Briefly, we observed good performance on real-world ECG images, similarly to the datasets we had created (**Table 3**). The similar performance of the models in both these settings suggests that our model has learnt generalizable information relevant for real-world application, with the ability to handle various types of noise in these data.

Comment 2: We believe also that adding a real-world ECG image dataset would greatly strengthen your case for publication.

Response: We appreciate the feedback from the Editors. We agree that a real-world image dataset is an important demonstration of the generalizability and performance of the model. To address this, we accessed two additional real-world datasets to test our models. Of note, these test data sets were not used during any model development and were only used in testing the final real world-adapted models (outlined in comment #1 above), to avoid any overfitting to real-world datasets.

There were two real-world datasets tested in our study:

- (1) **Rural US Hospital ECG dataset:** The represented a set of 64 ECGs from Lake Regional Hospital System (LRH) in Osage Beach, MO under a request for de-identified images we submitted to the hospital. These data included our 8-10 ECGs for our 6 labels of interest, as well as ECGs that were labelled as normal. All ECG labels were confirmed by a cardiologist. Subjectively, the ECG had a similar layout as the standard ECG format but had vertical lines demarcating the leads as well as had V1 rather than I as the rhythm

data. The signal was black rather than blue, and the background color and grid of the ECGs was slightly different, as was the location and font of lead label. These images are being submitted as Supplementary Information for review.

- (2) **Web-based ECG dataset**: In addition to the ECGs from Lake Regional Hospital, we also obtained 42 ECGs from the internet, as these would represent images not selected from any specific health system. We followed a systematic approach to constructing this dataset. We first accessed images on Life in the Fast Lane (LIFTL) website, an educational website for teaching ECGs available publicly on <https://litfl.com/ecg-library/>. From LIFTL, we took all ECGs for the labels of interest, without any pre-selection. Our goal was to have at least 6-7 ECGs per label. As the LIFTL had 2-6 ECGs per label, we pursued a google image search for “<label> ecg image” and selected the first 12-lead ECGs that appeared until we had 7 ECGs for each label of interest. All ECGs were confirmed by a cardiologist. The online set of ECGs included both standard and alternate format images, as well as novel format images such as ones with no rhythm leads, or two or more rhythm leads. The ECGs had varying background colors, signal colors, quality, and location of lead labels, and many had additional artifacts that were not present in our training data.

Based on the performance on the synthetic test set with noise including variation in color (Table R1), we did not expect the original model to be able to adapt to real-world ECGs, therefore, these assessments were only focused on the real-world adapted model presented in the study.

- For the Lake Regional Hospital dataset, as shown in **Table R4 (Table 3 in manuscript)**, the class mean weighted AUROC for the 64 images was 0.984, and the class mean weighted F1 score was 0.908.
- For the web-based ECG dataset, despite the contrasting nature of the images relative to all other data streams, our model had a performance similar to all other datasets (**Table R4**). Briefly, the class mean weighted AUROC for the images was 0.932, and the class mean weighted F1 score was 0.908. These images are being provided as Supplementary Information of review.

Table R4 / Table 3: Performance of Image Model on Real-World ECG Image Datasets

Dataset	Label	Accuracy	PPV	NPV	Specificity	Sensitivity	AUROC	F1	AUPRC
LRH	Male	0.719	0.667	0.842	0.516	0.909	0.778	0.769	0.784
	1dAVb	0.953	0.833	0.981	0.962	0.909	0.981	0.87	0.902
	RBBB	1.000	1.000	1.000	1.000	1.000	1.000	1.000	1.000
	LBBB	0.984	1.000	0.981	1.000	0.909	0.983	0.952	0.957
	SB	0.938	0.875	0.946	0.981	0.7	0.965	0.778	0.861
	AF	0.969	0.867	1.000	0.961	1.000	0.988	0.929	0.945
	ST	0.969	0.900	0.981	0.981	0.900	0.987	0.900	0.938
	Weighted Mean	0.969	0.912	0.983	0.980	0.909	0.984	0.908	0.935
LIFTL	1dAVb	0.884	0.625	0.943	0.917	0.714	0.81	0.667	0.679
	RBBB	0.930	0.700	1.000	0.917	1.000	0.944	0.824	0.693
	LBBB	0.930	0.833	0.946	0.972	0.714	0.897	0.769	0.763
	SB	0.977	0.875	1.000	0.972	1.000	0.992	0.933	0.962
	AF	0.977	0.875	1.000	0.972	1.000	0.988	0.933	0.938
	ST	0.953	0.778	1.000	0.944	1.000	0.960	0.875	0.761
	Weighted Mean	0.942	0.781	0.981	0.949	0.905	0.932	0.833	0.799

The manuscript has been updated as described below, with a new section in the results describing validation on real-world images (page 10, paragraph 2).

“We also pursued validation on two real-world image datasets. The first of these was from the Lake Regional Hospital System (LRH), a rural US hospital in Osage Beach, MO. These data included 64 ECG images including 8-10 ECGs for our 6 labels of interest, as well as ECGs that were labelled as normal. Subjectively, the ECGs had a similar layout as the standard ECG format but had the VI lead rather than lead I as the rhythm strip (a single lead with a full 10 second recording for identifying rhythm). There were vertical lines demarcating the leads, the signal was black rather than blue, and the background color and grid of the ECGs varied, as did the location and the font of lead label. Model performance on the LRH dataset was also comparable to the held-out test set. Class weighted mean AUROC and AUPRC were 0.98 and 0.94, respectively (Table 3). For gender, the model had an AUROC and AUPRC of 0.78.

The second real-world image dataset consisted of ECG images available on the internet, representing 42 ECGs. The approach to obtaining these images is outlined in the methods. All ECG labels were confirmed by a cardiologist. Qualitatively, these web ECGs included both standard and alternate format images, as well as new image formats such as ones with no rhythm leads, or two or more rhythm leads. Moreover, the ECGs had varying background colors, signal colors, quality, and location of lead labels, and many

had additional artifacts that were not present in our training data. These images are available upon request. The model achieved good performance on this web-based dataset, with class weighted mean AUROC and AUPRC of 0.93 and 0.80 respectively, and a high discrimination across labels (Table 3)."

We have updated the methods section to include more detailed descriptions of the real-world datasets (page 26, paragraph 2).

"We pursued validation on two real-world image datasets. The first of these was from the Lake Regional Hospital System (LRH), a rural US hospital in Osage Beach, MO. These data included 64 ECG images including 8-10 ECGs for our 6 labels of interest, as well as ECGs that were labelled as normal. Subjectively, the ECGs had a similar layout as the standard ECG format but had the V1 lead rather than lead I as the rhythm data. There were vertical lines demarcating the leads, the signal was black rather than blue, and the background color and grid of the ECGs varied, as did the location and the font of lead label."

"The second real-world image dataset consisted of ECG images available on the internet, representing 42 ECGs. We followed a systematic approach to constructing this dataset. We first accessed images on Life in the Fast Lane (LIFTL) website, an educational website for teaching about ECGs available at <https://litfl.com/ecg-library/>. From LIFTL, we took all ECGs for the labels of interest, without any pre-selection. Our goal was to have at least 6-7 ECGs per label. As LIFTL only had 2-6 ECGs per label, we pursued a google image search for "<label> ecg image" and selected the first 12-lead ECGs that appeared until we had 7 ECGs for each label of interest. Qualitatively, these web ECGs included both standard and alternate format images, as well as a new format of images such as ones without rhythm leads, or with two or more rhythm leads. Moreover, the ECGs had varying background colors, signal colors, quality, and location of lead labels, and many had additional artifacts that were not present in our training data. These images are available from the authors upon request."

We have also updated the discussion to include commentary on the performance on these real-world datasets (page 16, paragraph 3).

"We also found that the model performance was not limited to images generated from the waveform data but extended to those obtained as printed ECGs directly from different sources. The included ECGs spanned different colors, lead positions, and extraneous artifacts noted on subjective review of these data, but the model continued to have high discrimination, precision, and recall. These observations suggest the utility of both the real-world application of our model, but also provide a strategy for training ECG-based models on ECG images generated from current repositories of ECG signals and labels through the simulated introduction of real-world artifacts in the training data"

REVIEWER COMMENTS:

Reviewer #1:

Comment 1: Developing a model on printed ECG images rather than ECG signals addresses an important need in ECG automatic diagnosis, since signal-based method has difficulty in real-time and explainable applications, especially in remote environment. So it is important to explore the image-based model. An essential concern is that, image-based model is heavily dependent on the used ECG-image definition format. The authors should include the commonly used ECG-image formats. The reviewer suggests to include the introduction about the ECG-image standard and definition from different countries or representative device providers. The current version only includes two image format and their shuffled versions.

Response: We appreciate the feedback from the reviewer. We also recognize the need to develop an ECG image-based model that performs consistently across varying formats. Of note, the standard ECG format (four 2.5 second columns presented sequentially on the page, and a 10-second rhythm strip below) represents the most consistent 12-lead ECG format across the world and is a part of all ECG textbooks used by clinicians in learning to read ECGs.^{1,2} Based on the review of ECG data, and the experience of our team of cardiologists, real-world ECGs most commonly vary on the color of ECGs, location of lead labels, noise introduced by different start points, and rotations of images due to miscalibration of printing tools. Moreover, the number and lead(s) chosen for the rhythm strip (the continuous 10 second rhythm printed at the bottom of the 12-lead ECG) varies across ECGs.

As outlined in our response to the Editors' Comment #1, we accounted for differences in color, noise from different start points, and rotations by training both on a standardized subset of ECGs and a real-world variation subset. While the vast majority of ECG images are indeed in the standard format described above, to ensure the model learnt from the labels on ECG leads, rather than a single format, we explicitly added further formats of images, including an "alternate format" with two columns containing 5 second recordings. We also added variations of these "standard" and "alternate" formats, where leads were moved to atypical locations on the sheet. We validated and tested our models on these images in different formats. We recognize, however, that our generated test and external validation sets may not include formats and noise that are present in real-world printed ECGs.

To address this concern, we included a web-based real-world ECG test data that was derived from ECG images from online webpages (described in comment #2 by the Editors). These web-based ECGs included many formats not included in our training and validation data, including printed images with different leads displayed as the rhythm strip rather than lead I, with up to 3 rhythm strips reported. The details of these updates are included in our response to Editors' Comment #2 above.

Comment 2: Another concern is that, the authors involved a large amount of ECG data, however, the number of classification for ECG abnormalities was small (only 6). Thus, the practical value of the developed models is limited. As comparison, the

PhysioNet/Computing in Cardiology challenge 2020 involved 121 types of ECG abnormalities and challenged the classification task of 27 types (see <https://physionetchallenges.org/2020/>). The current classification task is relatively simple. The reviewer suggests to add more ECG abnormalities to improve the practical value of the model.

Response: We acknowledge that a model designed to detect all possible abnormalities would have the broadest clinical application. However, that was limited by 2 considerations. First, the broad training data from UFMG had labels for 6 abnormalities precluding assessment for other labels, Second, the key goal of the study was to demonstrate that image-based models could be trained for automated label detection and achieve performance similar to the published models developed on raw electrocardiographic signals. In addition, we now demonstrate that the introduction of artificial variations while generating training ECG images from signals also allows the model to be adaptable to real-world ECGs, with important implications for their intended application.

While we are unable to develop a broader model in this study, our goal is to evaluate additional labels through access to repositories with broader sets of labels. We have added this to our Discussion section (Page 18, Paragraph 2):

“Second, we focused on 6 clinical labels, based on their availability in the training data, and therefore, our models would not apply to other clinical disorders. We believe that our study represents a strategy of leveraging ECG images for a broad set of disparate diagnosis – spanning rhythm and conduction disorders, as well as hidden labels. We also demonstrate how existing data repositories with waveform data can be augmented to accomplish this task. Our goal for future investigations will be to design custom models on repositories with broader sets of labels as well as extract waveform-specific measures through access to valid information from richer data repositories.”

Comment 3: Further concern is that, how to validate the accuracies of ECG labels for all 2,228,236 12-lead ECGs? This question should be clearly clarified.

Response: We thank the reviewer for this comment. The reads for these 2,228,236 represent clinical evaluation by a cardiologist. Therefore, they represent the reads that were used in clinical care. While we believe these to represent the standard clinical practice of interpreting ECGs, it is conceivable that there are occasional diagnostic errors. However, we found that our model learnt despite the real-world nature of the data and the labels. Specifically, we found that high probability predictions initially noted to be false positives in both the held-out test set and the external validation dataset actually represented inaccurate clinical labels. In addition, the approach towards identifying the labels used in our current study, we have expanded the protocol for extracting labels from clinical reads, as described by Ribeiro et al.^{3,4} This has been added to the Data Source section of our Methods (Page 19, Paragraph 3):

“Briefly, labels for the primary CODE study dataset were obtained through the following procedure, as described by Ribeiro et al. Automated University of Glasgow statements and Minnesota codes obtained by the Uni-G automatic analysis software were compared to both automatic measurements provided by the Uni-G software and text labels extracted from expert reports written upon initial reading of the signals. These labels were extracted using a semi-supervised Lazy Associative Classifier trained on a dictionary created from text reports. Discrepancies in the labels provided by extracted expert annotation and automatic analysis were settled using both cutoffs related to ST, SB, and IdAVb, as well as manual review.”

Comment 4: Can the image-based model extract the ECG information like the channel (I, II, III, V1, etc.)? or ECG waveform features like QRS amplitude, hear rate? These explainable features outputs can enhance the model performance on multi-label classification tasks or more complicated tasks.

Response: We thank the reviewer for this comment. We however believe an extraction of these features, which can be done with wavelet detection algorithms, natively would not have much practical value. Moreover, incorporating such additional labels would be computationally challenging as it would require reweighing the learning strategy and loss functions, and best represent a distinct model and study. We agree that identification of many distinct features of the ECG will ultimately allow automation of the automated ECG reading. We have updated the limitations section as below (page 18, paragraph 2):

“Second, we focused on 6 clinical labels, based on their availability in the training data, and therefore, our models would not apply to other clinical disorders. We believe that our study identifies a strategy of leveraging ECG images for a broad set of disparate diagnosis – spanning rhythm and conduction disorders, as well as hidden labels. We also demonstrate how existing data repositories with waveform data can be augmented to accomplish this task. Our goal for future investigations will be to design custom models on repositories with a broader set of labels as well as extract waveform-specific measures through access to valid information from richer data repositories.”

Reviewer #2:

Comment 0: This is a high quality and interesting study by Sangha and colleagues focusing on the application of deep neural networks to ECG data to perform labeling of selected clinical arrhythmia and conduction disorder patterns. This is an area that has seen considerable attention in recent years, notably the studies from Hannun et al. (single lead digital signal, ambulatory ECG), Ribeiro et al (12-lead digital signal), and Gliner et al (12-lead digital signal, plus image-based model for atrial fibrillation), and this work extends these works in important ways. In particular, the potential of an image-based neural network for ECG rhythm detection is a primary focus of this work because it is able to process data in settings where access to the digital signals is not readily available. The authors present analyses within the large Clinical Outcomes in Digital Electrocardiology (CODE) dataset from Brazil, as well as an external dataset from Germany. As expected (based on the works Hannun and Ribeiro), the models demonstrate excellent performance in the multi-label classification task, including excellent performance in the independent test set and robustness to the specific format of the ECG image (which I thought was a particularly clever design element). What is surprising, however, is that not only was the image-based neural network comparable to the signal-based model (reported in the Gliner study), its performance appears to be generally superior. Moreover, the inclusion of Grad-CAM results helps to demonstrate that the models are cueing on clinically relevant features for classification (where human readers have an understanding of what cues to look for). Overall, I thought this was strong work, but do have some questions and suggestions for potential consideration and improvement.

Response: We thank the reviewer for their thoughtful comments. We have incorporated their comments, with changes to the analyses, and clarity of presentation. We have responded to their comments below.

Comment 1: There are several pre-processing steps described in the generation of the ECG images; for example, resampling to 300 Hz and starting the signal readout at a pre-determined time with respect to the QRS peak. I think clarifying the rationale for these steps is important (eg, are those considered standard pre-processing steps for ECG printing?) because otherwise, such manipulation runs counter to the premise of the work in seeking to make the compatibility of ECG data robust for neural network ingestion. In other words, are these images more idealized and regularized than what may be encountered in ‘the real world’?

Response: We thank the reviewer this insightful comment. We have now clarified this for the readers. The resampling to a set frequency is a standard preprocessing step done for ECG waveforms used in signal based models.^{3,5-8} The generation of ECG-images from the ecg-plot software was not affected by this resampling, as plotted images already represent down-sampling of the waveform to be displayed on the plot. We have included the clarification in the Data Preprocessing section of the Methods (page 20, paragraph 3, changes in boldface text):

“We resampled all ECGs to a 300 Hz sampling rate. Such a down-sampling of signals represents a standard preprocessing step for ECG waveform analyses to allow for standard data structure required for modeling.”

We also recognize that standardizing images of the plotted ECGs will reduce the variability that we are likely to encounter in the real-world ECG images. Therefore, we have revised our approach to plotting ECG images in our training sample. As described in our response to Editor Comment #1, we augmented the training sample with an additional 2,005,412 ECG images that represented variations in the background color and starting point for the original training data set. These additional ECGs had a starting point that was selected from a uniform distribution of 0 to 1.3 second offset from the first detected QRS complex in the signal. The background color and color of the signal trace was modified to include three different color variations (**Figure R1, Figure 1 in the manuscript**). This is in addition to the original training sample that only varied the layout of the leads but chose predetermined start point for plotting ECG images, at 300ms before the second detected QRS peaks. All of these transformations were stratified by clinical labels. Additionally, all 4 million training images were rotated a random amount between -10 degrees and +10 degrees during the model training process.

Furthermore, as mentioned in our response to the Editors’ Comment #1, we found that such a model was robust to noise-related artifacts anticipated in real world data, as well as in real-world printed ECG samples. (**Table R2, Supplementary Table 3**).

The changes corresponding to these comments have been included as excerpts in response to comments #1 and #2 above.

Comment 2. Similarly, since the study is focused on characterizing the performance of an image-based ECG deep learning model, I think it would be appropriate to add an additional sensitivity analysis with respect to varying image quality and/or noise characteristics, particularly given that scanned images are stated use case.

Response: We agree with the reviewer’s comments. As outlined in our response to comments #1 and #2 by the Editors and the comment #1 by the reviewer, we have addressed the performance of image-based models on datasets with real-world noisy artifacts as well as evaluated such models on samples of real-world ECGs. We demonstrate that image-based models can be trained to be generalizable, robust to noise, and interpretable. As mentioned in our response to Editor Comment #1, we generated 3 ECG image data subsets with three types of noise we anticipated in real-world images. Briefly, this included variations in the color of both the image background and printed data from the leads, rotations of the image, and random starting points for the waveforms on the printed images. As outlined in our response to the comment #1 and #2 by the Editors, the models trained on these augmented training data, were robust to noise in real-world data (**Table R2** above). This model also demonstrated impressive label discrimination on real-world printed ECGs, drawn from both a distinct US hospital as well as ECG images drawn from the internet, with images varying vastly on quality, format, and color (**Table R4** above).

The changes corresponding to these comments have been included as excerpts in response to comments #1 and #2 above.

Comment 3: It was a rather surprising finding to me that the image-based model outperformed the signal-based model, especially considering that the image-based model theoretically does not contain any extra information and the signal-based model even had additional inputs (median intervals, etc). Despite this surprising finding, there was not much consideration or speculation as to the source of these findings in the discussion. I think some additional discussion on this point is warranted.

Response: We thank the reviewer for this important point. We have now included additional points in the Discussion addressing these observations (Page 15, Paragraph 2).

“An important observation is that image-based models demonstrate comparable performance to our signal-based model, as well as signal-based models in published reports despite both the substantial down sampling of high frequency signal recordings to relatively low-resolution images, and the redundant information introduced by the presence of background pixels. We do not have a definitive explanation of these observations, though pixel-level information that is not interpretable by humans may include more detailed diagnostic clues than the review of broad waveform patterns used by human readers. Moreover, it is possible that while the signal-based models have a higher frequency record of electrocardiographic activity of the heart, and therefore, more data points that there is spatial information outside of waveform data that is better represented in printed images, or that the additional high frequency recording of noisier signal data does not necessarily have more information than image data. We cannot definitively prove if either of these is the case.”

Comment 4: I assume that the 827 ECGs in the Cardiologist-Validated test set matches the same 827 ECG test set from the Ribeiro study but, if so, this point should be made more explicitly. Also, if so, is the cardiologist reader from the Ribeiro study one of the 2-to-3 cardiologist readers for this analysis? Finally on this point – did the cardiologist reviewers have access to the prior clinical interpretations, or were their reviews blinded to any other interpretation?

Response: These are indeed the same 827 ECGs, and we have added this clarification in the revised manuscript, and the labels were the same as that study. The cardiologist readers in the Ribeiro et al were distinct from the readers who reviewed the web-based and real-world ECGs in the current study (RK and DJ), but followed the same best practices outlined in the AHA guideline for reading ECG used in the current study. Nevertheless, these reads were shared with Antonio Ribeiro, who is the PI of the CODE study and a coauthor on the current study. The cardiologist reviewers did not know the clinical label assigned to the ECG before their review.

The Methods section has been updated, as below (Page 20, Paragraph 2):

“This consisted of additional ECGs obtained from the TNMG network between April and September 2018. These ECGs were rigorously validated by 2-to-3 independent cardiologists based on criteria from the American Heart Association. This represents the cardiologist-validated test set from the study by Ribiero et al.”

Comment 5: With respect to the generation of ECG images with varying formats – were studies duplicated across formats within the held-out test set, or was each study only represented once?

Response: For the held-out test set assessments across different layouts, we represented each of the individual images in evaluating the two layout variations (Standard vs Alternate format images). This was done to ensure that model performance could be compared for the same images, only varying by the format. However, to ensure we were not duplicating the test set, we reported the performance of the model on the standard layout in **Table 1**, and present an assessment on the corresponding alternate layout in **Supplementary Table 4**. Similarly, the results for the cardiologist-validated test sets are presented in **Supplementary Table 5**. We note that it was not as clearly presented how the test set metrics were presented with respect to the ECG formats, and we have updated the results to highlight it more clearly. Changes to the manuscript are included below:

(Methods, Page 26, Paragraph 1)

“For the image-based models, the performance was evaluated separately for the both the standard and alternate ECG lead layouts of the held-out and the cardiologist validated test sets.”

Comment 6: In the paragraph describing the results for the held-out test set for the image data, I believe there is a typo in describing the specificity at maximum F1 (0.88 instead of, presumably, 0.98).

Response: Yes, we apologize for the typographical error. This has been corrected.

Comment 7: With respect to the confusion matrices in Figure 4, it is unclear how the reported ‘accuracy’ numbers are readily derived from the numbers presented in the figure and why they differ so drastically from those reported for the same labels in Table 1.

Response: We thank the reviewer for highlighting this. The predicted values and true labels axes labels were not appropriately labeled. We apologize for this, which have now been corrected below. These results are now concordant with Table 1. We thank the reviewer for their careful review.

Figure 4: Confusion Matrices for Image Model Predictions (next page)

Supplementary Figure 3: Confusion Matrices for Signal Model Predictions

Comment 8: There are no details provided for the peak annotation technique mentioned in the ‘Data Preprocessing’ section of the Methods.

Response: We have added some additional details about the peak annotation technique, as well as references upon which it is based. This has been added to our Methods section (Page 21, Paragraph 1):

“For all ECGs, we found the median and standard deviation of all RR, PR, QRS, and ST intervals for each lead using NeuroKit2, a Python toolkit for signal processing.⁹ **Briefly, the NeuroKit2 algorithm is based on the detection and delineation algorithms of Martinez et al.¹⁰, and uses a discrete wavelet transform to localize QRS peaks, allowing it to identify local maxima associated with these peaks regardless of noise artifacts that may be present in the signal. It then performs a guided search for P and T waves based on the information about QRS location and known morphologies for these waves in electrocardiographic signals. The median lengths of each interval across leads were included as additional inputs into the signal model.”**

Reviewer #3:

Comment 1: The paper describes applying two different types of convolutional neural networks (CNNs) on ECG images transformed from ECG signals and raw ECG signals respectively to identify 6 clinical labels as well as the subject gender. The results demonstrate that ECG images, similar as signals, can be used to classify the 6 clinical disorders and define hidden features for identifying the subject gender. Also, it shows the application of Gradient-weighted Class Activation Mapping (Grad-CAM) to highlight the regions in the image predicting a given label which provides interpretability.

The work is creative in a way that it implements an automated diagnosis model (with a deep neural network) for ECG images in addition to ECG signals. As the author mentioned in the paper, ECGs are frequently printed and scanned as images. However, the ECG images used in the study are generated from the ECG signals directly using the ecg-plot python library. The plotted ECG images are very different than the printed and scanned ECG images for the color, light, shadow and integrity sometimes, which makes the study less practical.

Response: We thank the reviewer for their comments. This is similar to the comment #1 by the Editors. We have explicitly revised the model development process to incorporate noise and artifacts in training. We describe these processes in detail in our response to the comment. Briefly, we have now included variation in color of ECG foreground and background, start points for plotting leads, and rotations of ECG images during training. In addition, we have including two real-world datasets for testing our model performance on real-world ECG images, not generated by us. The performance on these data have been described in our response to the Editors' Comment #2, where we include the performance on a set of ECGs from a rural US hospital (Lake Regional Hospital), and ECG images derived from the internet. Briefly, we demonstrate that our updated model performed well on both datasets. As outlined by the reviewer, the web-based image dataset in particular included particularly noisy artifacts, including varying background colors, signal colors, quality and size of image, and location of leads and their labels. Our models are robust to these variations.

The changes corresponding to these comments have been included as excerpts in response to comments #1 and #2 above.

Comment 2: The study is based on a dataset including 2,228,236 12-lead ECGs from 1,506,112 patients, which makes the results convincing. Moreover, the evaluation of the model also includes a secondary cardiologist-validated annotation test dataset that can assess the model trained from dataset that annotated from a single clinician. In addition, an external validation set collected from another country avoids the doubts about bias caused by the population distribution. The number of ECGs with at least one of the six disorders (231,704) is a small amount compared to the whole dataset, which may affect the model since it sees much more normal cases, and the effect is better to be explored and discussed. Similarly, the ECGs with the six clinical labels (AF, RBBB, LBBB, 1dAVb, SB and ST) are

skewed with more RBBB (61,551) and SB (48,296) than others (from 34,446 to 39,661), while there is no discussion about the effects of the skewness in data.

Response: We thank the reviewer for raising this important point. Class imbalances in the training set represent a key issue that were explicitly addressed when training both image and signal models. These were important to ensure that the model had adequate sensitivity to each of the six clinical labels as well as the label for gender, and did not merely maximize performance training on the more common labels. To address this, we used custom loss functions with an Effective Weighting scheme developed for long tailed imbalanced datasets. We have clarified this in the Experimental Design section of the Methods (Page 23, Paragraph 2)

“To ensure that model learning was not affected by the low frequency of certain labels, custom loss functions based on the effective number of samples class sampling scheme were used for both image and signal models, with weighting based on the number of samples for each class (Supplementary Table 2). **We gave higher weights to rarer classes with the goal of ensuring that performance on metrics sensitive to class imbalances remained high.**”

Additionally, we included metrics such as the area-under the precision recall curve (AUPRC) and the F1 score that are more sensitive to performance of rare labels than traditional metrics like accuracy and AUROC.¹¹ We demonstrate impressive performance on these metrics, and have updated the Statistical Analysis subsection of the Methods to clarify this (Page 28, Paragraph 4)

“We assessed area under the receiver operating characteristic (AUROC) curve, which represents model discrimination, with values ranging from 0.50 to 1.00, representing random classification and perfect discrimination of labels, respectively. **In addition, we assessed the area under the precision recall curve (AUPRC), and F1 score of the model for each label, metrics which are sensitive to rare events and may provide more insight on the clinical utility of our models.**¹¹⁻¹³ We also assessed the sensitivity, specificity, positive predictive value, and negative predictive value for each label.”

Comment 3: The paper brings some interesting points for processing the inputs such as using different leads layout in the image to ensure that “the model learned about lead-specific information based on the label of the lead, rather than just the location of leads on a single ECG format”. It also applies the Grad-CAM method to identify the ECG regions that are most important for the label classification. However, it does not have enough analysis on whether or not using the different image formats achieve the goal of preventing the model from just learning the location of leads on a single ECG format.

Response: We thank the reviewer for this comment. In our original analyses, we included information from standard and alternate layouts of the held-out and validation test sets and demonstrate similar performance. We include these as Supplementary Tables 4 and 5, but agree that we hadn’t called them out specifically. We have now revised this, as below:

(Results, Page 14, Paragraph 4)

“Our image-based model performed comparably on both standard and alternate form printed images in the held-out test set and cardiologist validated test set (Supplementary Tables 4 and 5).”

(Methods, Page 26, Paragraph 1)

“For the image-based models, the performance was evaluated separately for the both the standard and alternate ECG lead layouts of the held-out and the cardiologist validated test sets.”

The Grad-CAMs, in addition, demonstrated that the model had not only learned one specific location for leads and was able to incorporate images in both standard and alternate formats. However, we acknowledge the need to evaluate that the generalizability extends to additional formats that were not explicitly present in the training set.

As mentioned in our response to the comment #1 by the reviewer, we have further addressed this in our presentation of our revised model that we tested on real-world ECG images. Specifically, on real-world images, with formats that varied beyond what was included in the training data, we still observed high performance, and GradCAM still identified areas that were lead-specific and clinically relevant when making its prediction. We included two examples of both RBBB and LBBB in the new Supplementary Figure 4 for these individual real-world examples and updated the Results section (Page 14, Paragraph 2). The figure is also reproduced below.

“Supplementary Figure 4 shows Grad-CAMs for individual representative examples of model prediction of RBBB and LBBB on real-world images from the web-based dataset. In both examples of RBBB, the region corresponding to leads V1 and V2 is most important for prediction of the label. In the two examples of LBBB, precordial leads are again the most important for prediction of the label, despite varying in the relative position of the leads and the difference in the number and type of the continuous rhythm strip at the bottom of the ECG image.”

Supplemental Figure 4. Gradient Class Activation Maps (Grad-CAMs) for web-based real-world printed electrocardiograms for right and left bundle branch block (RBBB, and LBBB).

A. RBBB

B. RBBB

C. LBBB

D. LBBB

We have also included the following commentary to summarize these points in the Discussion section (Page 17, Paragraph 1, changes in boldface text).

“Our examination of mean heatmaps across a sample of predictions for RBBB and LBBB demonstrates that classifications for these intraventricular conduction disturbances were guided by the same information human readers focus on when reading an ECG. **Mean heatmaps consistently demonstrated the identification of specific leads that are important in clinical diagnosis of RBBB and LBBB.** Similar heatmaps applied to individual ECGs further supported a similar interpretable learning across clinical labels. **The identification of specific leads despite the variety of inputs in the training data, and the rotation of training images suggests that model has learned more generalizable representations of ECG images, especially as it still identified clinically relevant leads in formats of images in the web-based real world image dataset, which it had never encountered in model training.**”

Comment 4: The paper emphasizes the development of “dual modality multilabel prediction algorithms that incorporate either ECG images or signals as inputs.” However, it does not provide any integration mechanism to coordinate the image classification model which is based on an Efficientnet-B3 architecture, and the signal classification model which

is based on an inception architecture. In addition, the paper does not include studies considering the characteristics about ECG images and/or ECG signals, and lacks analysis on model selections and comparisons.

Response: We thank the reviewer for this comment. As mentioned in response to Editor Comment #1, the primary purpose of the models was to demonstrate the effectiveness of a model that was trained on either images or signals for automated clinical diagnosis and hidden label detection, while demonstrating the generalizability and interpretability of such an approach. A key advance of our work is the focus on ECG images and defining strategies to train real-world image-adapted models from waveform data.

Therefore, our approach emphasizes the design and testing of ECG-image based models. The signal-based models have been deployed for many applications in ECG, and we included a signal model to allow a more direct comparison of the model performance of the two modalities. In terms of model selection for signals, as mentioned in the manuscript, we adapted an architecture from Raghunath et al. that had been published recently and had demonstrated promising results identifying new-onset atrial fibrillation.

Of note, since the source data in a real-world setting would either be a signal or an image, the different model architectures do not pose an issue in its application. We did not include rigorous comparisons of these trials as we felt that they were not central to the focus of the paper as the signal model was simply meant to be a benchmark for the performance of any image-based models.

We have removed reference to a dual modality multilabel prediction system, and more clearly stated our goals. We have made additional changes throughout the manuscript to highlight our focus including excerpts in our response to Comment #1 and #2 by the Editors and Comment #3 by Reviewer #2.

Comment 5: Overall, the study shows promising results on using an Efficientnet-B3 network to classify 6 clinical disorders as well as genders from ECG images generated from ECG signals. It is based on a valuable large ECG dataset as well as distinct and external evaluation sets to show the generalization of the model. However, because the plotted ECG images are different than the printed ones, it reduces the ability of using the model for the real-world applications. Also, it lacks analysis on the skewness in data and the effects of different layouts for the input image. Moreover, it does not show any integration mechanism for the two models of the dual modality multilabel prediction system.

Response: We appreciate this comment and others from the reviewer. Their suggestions have been incorporated in our response to comments #1 and #2 by the Editors and in response to comment #3 by the reviewer. These suggestions were incorporated as additional analyses as well as changes to the Discussion in the manuscript, which have strengthened the paper. While we have described in detail in our response to the referenced comments, briefly, we addressed the concerns about the lack of skewness and other real-world noise artifacts in both our training and test data by training a new real-world adapted model, which included training on images with different background colors and shifted starting points as well as random rotation. We confirmed that this real-world model did not have a drop-off in performance on various noise artifacts we

artificially added to a subset of the held-out test set (Table R2), Please see our response to Editor comment #1 for additional details on the training of this new real-world model.

We have addressed concerns about the differences between plotted ECG images and real-world printed ones in our response to Editor comment #2. To briefly summarize, we tested our model on two real-world datasets. The first of these was an ECG dataset obtained from a rural US hospital system whose ECGs had a similar layout as the standard ECG format but had additional artifacts and differences in the color of both the lead waveforms and background. The second of these was a web-based ECG dataset, which included both standard and alternate format images, as well as novel format images such as ones with no rhythm leads, or two or more rhythm leads. The ECGs had varying background colors, signal colors, quality, and location of lead labels, and many had additional artifacts that were not present in our training data. We saw similar performance on these real-world datasets as we did on our plotted ECG ones. The changes corresponding to this comment are included in excerpts in our response to Comment #2 by the Editors. And finally, as outlined our response to Comment #3 by the reviewer, we demonstrate that the real-world ECG image datasets had interpretable performance on evaluation of model predictions using gradient-class activation maps (Grad-CAMs). We are sincerely appreciative of the reviewer for their constructive and thoughtful comments.

Response References

- 1 Hampton, J. R. *The ECG made easy*. 4th edn, (Churchill Livingstone, 1992).
- 2 O'Keefe, J. H. *The ECG criteria book*. 2nd edn, (Physicians' Press, 2010).
- 3 Ribeiro, A. H. *et al.* Automatic diagnosis of the 12-lead ECG using a deep neural network. *Nat Commun* **11**, 1760, doi:10.1038/s41467-020-15432-4 (2020).
- 4 Ribeiro, A. L. P. *et al.* Tele-electrocardiography and bigdata: The CODE (Clinical Outcomes in Digital Electrocardiography) study. *J Electrocardiol* **57S**, S75-S78, doi:10.1016/j.jelectrocard.2019.09.008 (2019).
- 5 Hughes, J. W. *et al.* Performance of a Convolutional Neural Network and Explainability Technique for 12-Lead Electrocardiogram Interpretation. *JAMA Cardiol*, doi:10.1001/jamacardio.2021.2746 (2021).
- 6 Hannun, A. Y. *et al.* Cardiologist-level arrhythmia detection and classification in ambulatory electrocardiograms using a deep neural network. *Nat Med* **25**, 65-69, doi:10.1038/s41591-018-0268-3 (2019).
- 7 Raghunath, S. *et al.* Deep Neural Networks Can Predict New-Onset Atrial Fibrillation From the 12-Lead ECG and Help Identify Those at Risk of Atrial Fibrillation-Related Stroke. *Circulation* **143**, 1287-1298, doi:10.1161/CIRCULATIONAHA.120.047829 (2021).
- 8 Biton, S. *et al.* Atrial fibrillation risk prediction from the 12-lead ECG using digital biomarkers and deep representation learning. *European Heart Journal - Digital Health*, doi:10.1093/ehjdh/ztab071 (2021).
- 9 Makowski, D. *et al.* NeuroKit2: A Python toolbox for neurophysiological signal processing. *Behav Res Methods*, doi:10.3758/s13428-020-01516-y (2021).
- 10 Martinez, J. P., Almeida, R., Olmos, S., Rocha, A. P. & Laguna, P. A wavelet-based ECG delineator: evaluation on standard databases. *IEEE Trans Biomed Eng* **51**, 570-581, doi:10.1109/TBME.2003.821031 (2004).
- 11 Stevens, L. M., Mortazavi, B. J., Deo, R. C., Curtis, L. & Kao, D. P. Recommendations for Reporting Machine Learning Analyses in Clinical Research. *Circ Cardiovasc Qual Outcomes* **13**, e006556, doi:10.1161/CIRCOUTCOMES.120.006556 (2020).
- 12 Ozenne, B., Subtil, F. & Maucort-Boulch, D. The precision--recall curve overcame the optimism of the receiver operating characteristic curve in rare diseases. *J Clin Epidemiol* **68**, 855-859, doi:10.1016/j.jclinepi.2015.02.010 (2015).
- 13 Leisman, D. E. Rare Events in the ICU: An Emerging Challenge in Classification and Prediction. *Crit Care Med* **46**, 418-424, doi:10.1097/CCM.0000000000002943 (2018).

Reviewers' Comments:

Reviewer #2:

Remarks to the Author:

The authors have been very responsive to the prior critiques. Congratulations on this great work.

Reviewer #3:

Remarks to the Author:

Thanks for the authors' responses. My concerns have been addressed well, and I have no further comments.